# Controlling spatiotemporal pattern formation in a concentration gradient with a synthetic toggle switch

Içvara Barbier[1] iD, Rubén Perez-Carrasco[2,3,*] iD & Yolanda Schaerli[1,**] iD

## Abstract

The formation of spatiotemporal patterns of gene expression is frequently guided by gradients of diffusible signaling molecules. The toggle switch subnetwork, composed of two cross-repressing transcription factors, is a common component of gene regulatory networks in charge of patterning, converting the continuous information provided by the gradient into discrete abutting stripes of gene expression. We present a synthetic biology framework to understand and characterize the spatiotemporal patterning properties of the toggle switch. To this end, we built a synthetic toggle switch controllable by diffusible molecules in *Escherichia coli*. We analyzed the patterning capabilities of the circuit by combining quantitative measurements with a mathematical reconstruction of the underlying dynamical system. The toggle switch can produce robust patterns with sharp boundaries, governed by bistability and hysteresis. We further demonstrate how the hysteresis, position, timing, and precision of the boundary can be controlled, highlighting the dynamical flexibility of the circuit.

**Keywords** bistability; dynamical systems; gene regulatory networks; pattern formation; synthetic biology

**Subject Categories** Biotechnology & Synthetic Biology; Computational Biology; Microbiology, Virology & Host Pathogen Interaction

**Mol Syst Biol. (2020) 16: e9361**

## Introduction

Synthetic biology aims to engineer living organisms with standardized and modular circuits that perform their functions in a programmable and predictable way (Brophy & Voigt, 2014; Cameron *et al*, 2014; Purcell & Lu, 2014). In addition to the promise of providing new technologies for medical and industrial applications (Nielsen & Keasling, 2016; Gilbert & Ellis, 2018; Kitada *et al*, 2018; Xie & Fussenegger, 2018), recapitulating biological processes synthetically

provides a route to understand the basic necessary mechanisms underpinning biological functions and dissect their properties and limitations (Bashor & Collins, 2018; Li *et al*, 2018).

Formation of spatiotemporal patterns of gene expression, a crucial process during the development of multicellular organisms, lends itself to be studied by such a synthetic biology approach. During development, pattern formation is achieved through a set of inter-connected gene regulatory programs encoding different non-linear responses to spatial chemical cues. This multiscale complexity makes the elucidation of the core principles of spatial patterning very challenging in living embryos, calling for alternative approaches capable of interrogating and comparing different pattern formation mechanisms. The rise of synthetic biology has successfully allowed to build synthetic systems able to explore core patterning principles (reviewed in Davies (2017), Ebrahimkhani and Ebisuya (2019), Luo *et al* (2019), Santos-Moreno and Schaerli (2019b)). In addition, synthetic pattern formation is also an attractive technology for the engineering of living materials (Cao *et al*, 2017; Gilbert & Ellis, 2018; Nguyen *et al*, 2018; Moser *et al*, 2019) and tissues (Davies & Cachat, 2016; Webster *et al*, 2016; Healy & Deans, 2019).

One ubiquitous strategy of patterning during embryogenesis is positional information, in which signaling molecules—the morphogens—diffuse and generate concentration gradients. Specific gene regulatory programs are able to translate the spatiotemporal information provided by the local concentration of morphogen gradients into robust gene expression patterns (Wolpert, 1996; Rogers & Schier, 2011; Green & Sharpe, 2015). This mechanism has been extensively used in synthetic systems to generate spatial patterns, especially stripe patterns, which were produced through synthetic feed-forward loops (Basu *et al*, 2005; Schaerli *et al*, 2014), inducible promoters (Grant *et al*, 2016), and AND gates (Boehm *et al*, 2018).

One of the gene regulatory subnetworks able of interpreting positional information is the bistable genetic switch (Kraut & Levine, 1991; Alon, 2007; Lopes *et al*, 2008; Balaskas *et al*, 2012; Sokolowski *et al*, 2012; Zhang *et al*, 2012; Srinivasan *et al*, 2014; Perez-Carrasco *et al*, 2016; Zagorski *et al*, 2017), known as toggle switch (TS; Fig 1A). The topology of this circuit consists of two

1 Department of Fundamental Microbiology, University of Lausanne, Lausanne, Switzerland
2 Department of Life Sciences, Imperial College London, South Kensington Campus, London, UK
3 Department of Mathematics, University College London, London, UK
  *Corresponding author. Tel: +44 (0)20 7594 5114; E-mail: r.perez-carrasco@imperial.ac.uk
  **Corresponding author. Tel: +41 (0)21 692 56 02; E-mail: yolanda.schaerli@unil.ch

cross-repressing nodes that result in the binary mutually exclusive stable expression of one of the nodes. If the expression is influenced by an external signal, the TS provides a mechanism to convert a concentration gradient of this signal into stripes of gene expression (Sokolowski *et al*, 2012; Perez-Carrasco *et al*, 2016). Examples of TS-controlled pattern formation have been identified in the mesoderm formation in *Xenopus* (Saka & Smith, 2007), *Drosophila* blastoderm gap gene segmentation (Clark, 2017; Verd *et al*, 2019), and neural specification in vertebrate neural tube (Briscoe & Small, 2015; Perez-Carrasco *et al*, 2018).

The non-linearity of gene regulatory networks such as the TS impedes to intuitively understand the effect that different kinetic parameters have on the dynamics of gene expression. For this reason, during the last decade, gene regulatory networks have been analyzed using tools from dynamical systems theory, associating the stable steady states of the dynamical system with the attainable gene expression states of a cell. Similarly, the change in the availability of cellular states as a consequence of perturbations of kinetic parameters of the network can be associated with the bifurcations of the dynamical system, providing information of the constraints of

the possible cellular states. Indeed, the dynamical system of the TS has been thoroughly analyzed *in silico,* both in single cells and in population-level patterning scenarios (Ferrell, 2002; Guantes & Poyatos, 2008; Perez-Carrasco *et al*, 2016), showing that two possible stable states can coexist for a region of parameters inside which the expression state of the cell will depend on the initial gene expression—a phenomenon known as bistability. Under the control of an external signal, this bistability leads to hysteresis in which the state of the system is robust to signal changes, thus providing memory to gene expression patterns (Wang *et al*, 2009). Interestingly, the analysis of the steady states of the dynamical system provides not only static information of the cellular states, but also information on the transient dynamics of gene expression by which the states are attained (Verd *et al*, 2014). Therefore, a map of the underlying dynamical system is critical to fully understand the dynamics of the TS network.

The first synthetic version of the TS network was built almost 20 years ago and was a milestone of synthetic biology (Gardner *et al*, 2000). Since then, it has been built multiple times, extensively studied, and used for its memory, bistability, or hysteresis properties

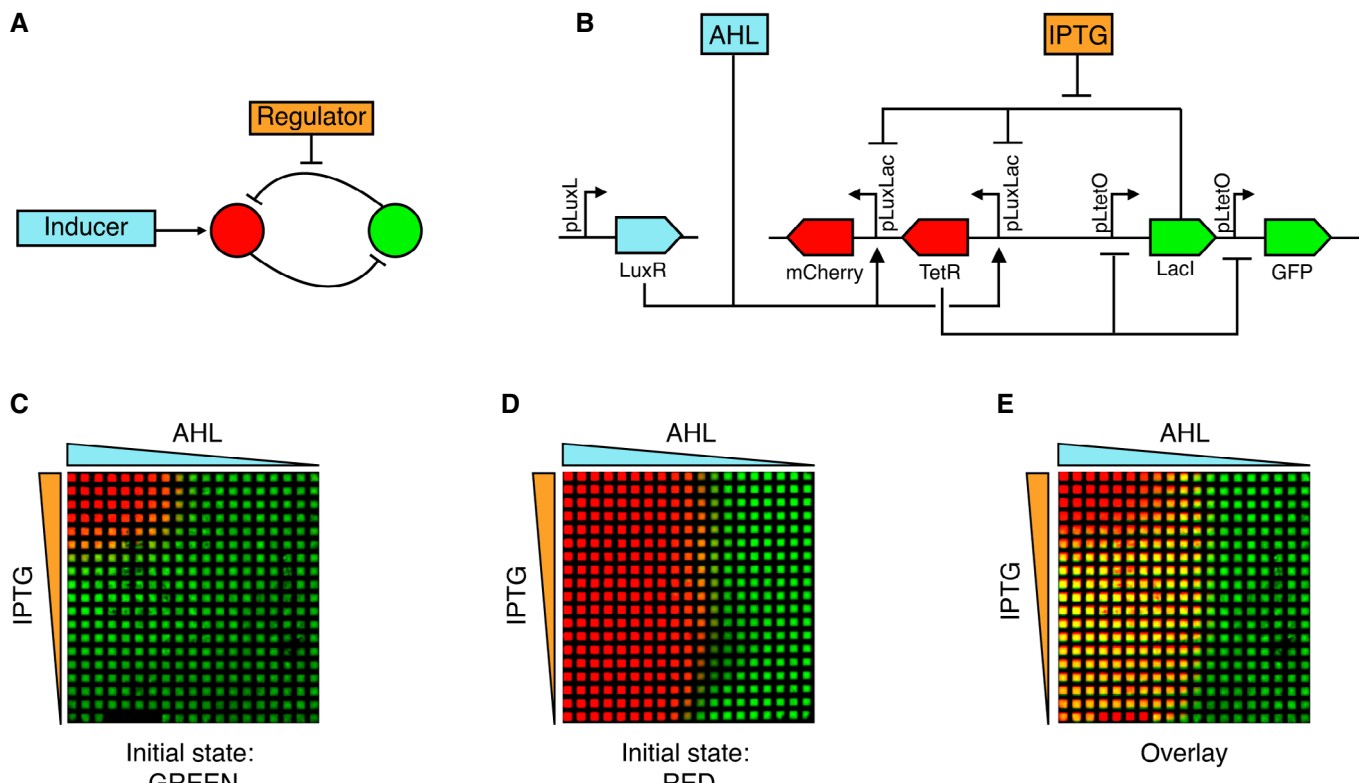

**Figure 1. Inducible toggle switch topology and its spatial patterning behavior.**

A  Schematic of the inducible toggle switch network composed of two mutually inhibitory nodes. The expression of the red node is controlled by an inducer, while a regulatory molecule can be added to tune the repression strength of the green node.

B  Detailed representation of the molecular implementation of the network in (A) using SBOL annotation (Beal *et al*, 2019).

C, D  2D spatial patterning of a population of *Escherichia coli* cells harboring the inducible toggle switch network. The colors correspond to the levels of fluorescence of mCherry (red) or GFP (green) produced by bacteria grown on a grid. Five microliter of aqueous solutions of 100 mM IPTG and 100 μM AHL was added at the grid edges forming gradients by diffusion as indicated by the triangular shapes. Before growing on the grid, bacteria were turned into the green (C) or red state (D) as indicated at the bottom. The grids were imaged after overnight incubation. Each grid is composed of squares with dimensions 0.75 × 0.75 mm².

E  Overlay of the grids of (C and D), highlighting the hysteresis of the system in yellow, resulting from the superposition of red and green fluorescence.

Source data are available online for this figure.

(Kim *et al*, 2006; Lou *et al*, 2010; Chen & Arkin, 2012; Padirac *et al*, 2012; Sokolowski *et al*, 2012; Lebar *et al*, 2014; Purcell & Lu, 2014; Zhao *et al*, 2015; Nikolaev & Sontag, 2016; Andrews *et al*, 2018; Pokhilko *et al*, 2018; Yang *et al*, 2019), for stochasticity fate choice (Sekine *et al*, 2011; Wu *et al*, 2013; Axelrod *et al*, 2015; Perez-Carrasco *et al*, 2016; Weber & Buceta, 2016; Lugagne *et al*, 2017), and to tune threshold activation (Gao *et al*, 2018). Nevertheless, its patterning capabilities controlled with a morphogen-like signal have not been studied in a synthetic system. Here, we constructed a "morphogen"-inducible synthetic TS network and studied its ability to produce spatial patterns—governed by bistability and hysteresis—in an *Escherichia coli* (*E. coli*) population. A combination of experiments and mathematical modeling allowed us to characterize the underlying bifurcation diagram, unveiling the possible dynamical regimes of the circuit. This enabled us to demonstrate how the inducible TS allows to control hysteresis, precision, position, and timing of the pattern boundary.

## Results

### Inducible toggle switch topology and its spatial patterning behavior

The inducible TS network consists of two mutually repressing nodes (Fig 1A). This mutual inhibition architecture ensures that only one of the nodes can be maintained at high expression. An array of cells under a concentration gradient in charge of controlling either the repression strength or the production rate of one of the nodes will generate a binary spatial pattern. We built a synthetic inducible TS circuit starting from a characterized TS (Litcofsky *et al*, 2012). The first node of the network is composed of the TetR repressor and the mCherry reporter and will be referred hereon as the red node. The second node contains the LacI repressor and GFP reporter and will be referred to as green node (Fig 1B). Accordingly, the two expression states that the circuit can maintain will be referred to as green and red states. TetR and mCherry are regulated by the hybrid pLuxLac promoter (BBa_I751502), which is activated by the LuxR-AHL (*N*-(β-ketocaproyl)-L-homoserine lactone) complex and repressed by LacI, whose repression strength can be regulated by isopropyl β-D-1-thiogalactopyranoside (IPTG). LacI and GFP are controlled by the pLtetO promoter, which is repressed by TetR. LuxR is constitutively expressed from a pLuxL promoter on a second plasmid (Fig 1B). In the absence of AHL (inducer), TetR and mCherry are not expressed, but LacI and GFP are, resulting in the green state. In the presence of AHL, TetR and mCherry can be expressed and repress LacI and GFP expression. Consequently, the system switches to the red state, provided that the concentration of the regulator (IPTG) is high enough.

We studied the capability of the inducible TS circuit to pattern an *E. coli* population exposed to different concentrations of AHL and IPTG and combinations thereof (Fig 1C and D). To this end, we grew the cells on a hydrophobic grid placed on an agar plate to give a defined spatial organization to the cells (Grant *et al*, 2016), while small molecules can freely diffuse between grid squares. AHL and IPTG were pipetted at the left and at the top edges of the grid, respectively, forming gradients of AHL and IPTG by diffusion, thus inducing a fluorescent pattern in the *E. coli* populations. We

measured the bacterial fluorescence after overnight incubation (~ 16 h). Although the gradients decay over time, the observed patterns stayed constant even after further incubation, as the expression of the synthetic toggle switch becomes frozen in cells that have entered the stationary phase, a phenomenon commonly observed for bacterial synthetic circuits (Elowitz & Leibler, 2000). When the starting cells were in the green state (reached by previous incubation in the absence of AHL and IPTG), the switch to the red state was observed only in the presence of both AHL and IPTG, in the top left corner of the grid (Fig 1C). In contrast, the same experiment performed with cells initially in the red state (reached by previous incubation in the presence of AHL and IPTG) showed that the red state is maintained above a certain concentration of AHL, mostly independent of the concentration of IPTG, leading to a green domain at the right (Fig 1D). An overlay of the two grid patterns allows to highlight the possible stable states available for different concentrations of AHL and IPTG showing two monostable regions for the red and green states, and a bistable region for high values of AHL and low values of IPTG that grant hysteresis to the system (Fig 1E). In addition to the tuning of the LacI repression by IPTG as explored here, the TetR repressing strength can be regulated by the addition of anhydrotetracycline (aTc; Fig EV1A and B). Different combinations of gradients of AHL, IPTG, and aTc result in different patterns (Fig EV1B–D), revealing the flexibility of the TS to control spatial gene expression. We focused on AHL-IPTG because we aimed to have a TS which interprets a "morphogen" gradient (AHL) that acts on the expression level of the target by direct promoter interaction, similar to known morphogens such as bicoid (Chen *et al*, 2012) or sonic hedgehog (Balaskas *et al*, 2012). Moreover, this combination allowed us to observe a cusp bifurcation as we will describe below.

### Characterization of the toggle switch as a dynamical system

To quantitatively analyze the network behavior with single-cell resolution, we grew the TS bacteria in liquid culture for 10 h with defined concentrations of inducer (AHL) and regulator (IPTG) molecules. We prevented the bacteria from entering the stationary phase by diluting them 100-fold after 5 h. We used flow cytometry to measure the green and red fluorescence of individual bacteria (Fig 2A). We set gates for the red and green states with the help of positive red and green fluorescent controls. This gating allowed us to quantify the percentage of TS bacteria in the red or green state for different inducer and regulator concentrations and different initial states. The observed behavior is consistent with the spatial patterning on the grid: Starting with an initial population of bacteria in the green state, the entire population (> 90% of events) switched to the red state at high concentrations of both AHL and IPTG, whereas they stayed green otherwise. On the other hand, starting in the red state, the entire population stayed red above AHL concentrations of ~ 11 nM, but switched to the green state at lower AHL concentrations.

At the boundary between the red and green areas (white squares in Fig 2A), we observed cells in both flow cytometry gates. At high IPTG concentrations (≥ 0.5 mM), a single population was located at intermediate red and green fluorescence values, whereas at IPTG concentrations ≤ 0.5 mM, the population split into two subpopulations displaying either the red or the green state. A bimodal

distribution at the boundary between the two stable states is a known phenomenon of bistable circuits (Sekine *et al*, 2011; Wu *et al*, 2013; Axelrod *et al*, 2015) and can be caused by cell state switching in response to intrinsic gene expression noise (Sekine *et al*, 2011).

The transition from red to green is less affected by the concentrations of IPTG than the transition from green to red, because of the asymmetry of the network. In the transition from red to green, the rate-limiting step is the degradation and dilution of TetR in the cells, since the clearance of TetR is required before changes in

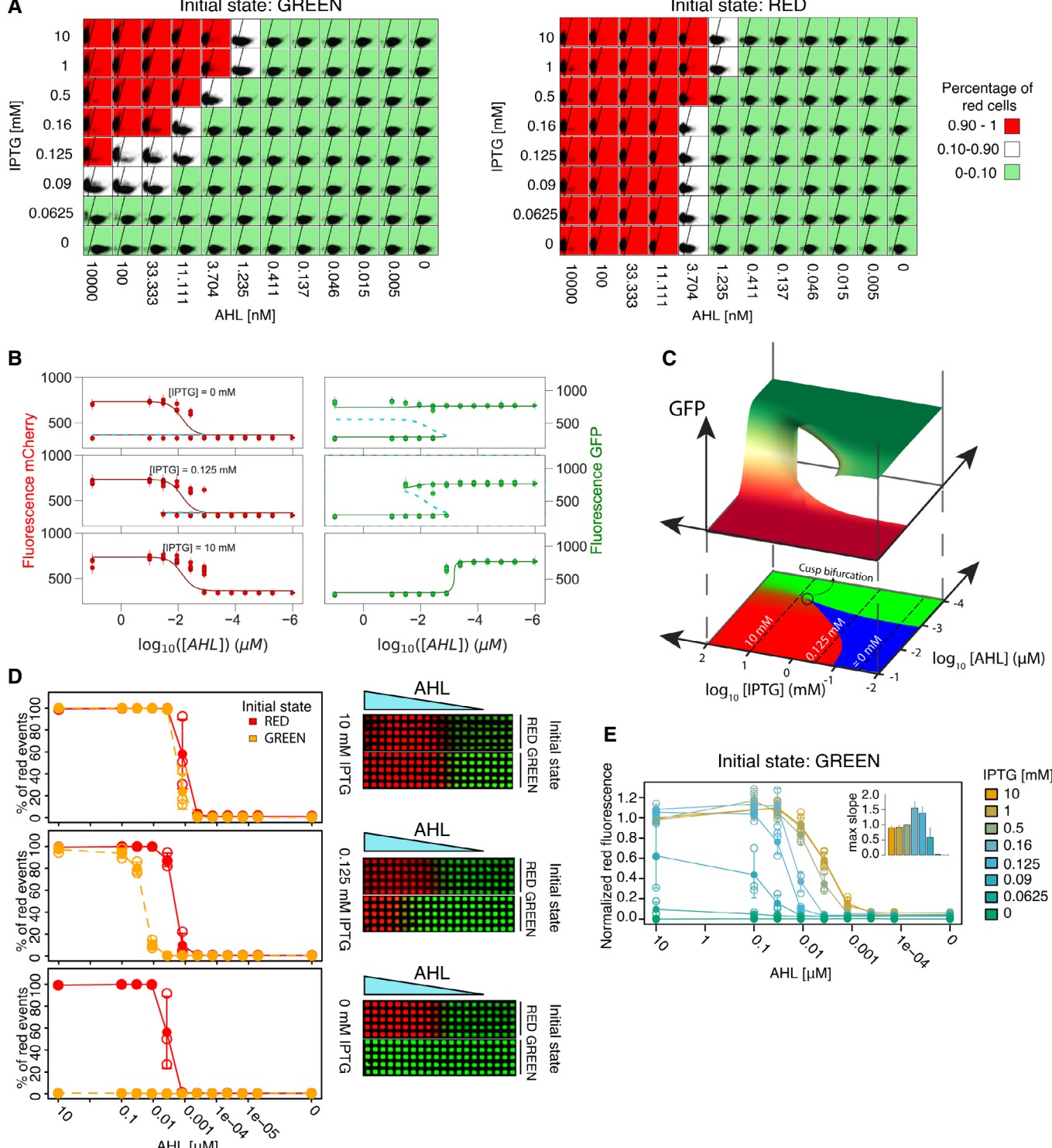

Figure 2.

**Figure 2. Controlling the hysteresis, position, and sharpness of the boundary.**

A   Quantitative single-cell analysis of the inducible toggle switch. Overview of the flow cytometry data of a representative replicate. Each square shows red (*y*-axis) and green (*x*-axis) fluorescence of the population (10,000 events) measured at indicated concentrations of IPTG and AHL after 10 h of incubation. Background color corresponds to the percentage of red gated events as indicated.

B   Comparison between the observed populated states from the flow cytometry data (circles) and the available steady states predicted by the model (solid lines: stable states; dashed lines: unstable states), shown for three IPTG concentrations. The median and the standard deviation of the experimentally observed states of the gated populations from three replicates are shown. Parameters used in the model are the best parameter candidates from the MCMC fitting (Table 1). The whole dataset is shown in Fig EV3.

C   Top: Bifurcation diagram for the parameters found in the fitting, showing the stable steady states available for different combinations of [AHL] and [IPTG]. Bottom: phase portrait of the TS, indicating monostable regions in red/green proportionally to the concentrations of mCherry/GFP and bistable conditions in blue. The circle indicates the cusp bifurcation.

D   Effect of initial conditions in the patterning of the TS showing the hysteresis at three different IPTG concentrations, corresponding to the three different regimes showed in (C) as dashed lines. Left: Mean percentage of red cells (full circles) for different concentrations of AHL are shown for an initial population in the red state (solid red lines) and for an initial population in the green state (dashed orange lines). Error bars show standard deviation of three biological replicates (individually shown as empty circles). Right: Grid patterns at constant IPTG concentrations. Five microliter of a solution of 100 µM AHL was added at the left to create the gradient. Colors correspond to the intensity of red and green fluorescence.

E   Sharpness of the boundary. Normalized red fluorescence for an initial population in the green state in an AHL gradient at different IPTG concentrations. Mean (full circles) and standard deviation (error bars) of three biological replicates (individually shown as empty circles). The inset bar plots represent the mean of the maximum slope of each curve from three replicates with standard deviation as error bars (Appendix Fig S1).

Source data are available online for this figure.

AHL or IPTG concentrations can induce the switch. Therefore, for the rest of the manuscript, we will focus on initial green state populations, while the data for the initial red state are in the Appendix Figs S1 and S2.

In order to fully characterize the dynamical capabilities of the circuit, we described the network with a mathematical model composed of ordinary differential equations (ODE) capturing the concentration of all the chemical species over time (see Materials and Methods for details). To unveil the dynamical landscape compatible with the synthetic circuit, we parameterized the model by fitting it to our experimental data. Since the transient expression profiles to both cellular states were faster than the transitions between them, we inferred the details of the dynamical system by identifying each observed cellular state with a stable steady state of the dynamical system. In order to do so, we made use of multitry Markov chain Monte Carlo (MCMC) inference (Laloy & Vrugt, 2012; Shockley *et al*, 2018), with a likelihood function based on the experimentally measured levels of expression and number of steady states found for the different concentrations of AHL and IPTG tested. We obtained a narrow probability distribution for all 11 parameters (Fig EV2). The predicted outputs from the parameterized equation were able to recapitulate the bistability and hysteresis observed in the experimental data (Figs 2B and EV3). This allowed us to reconstruct the multidimensional bifurcation diagram of the system (Fig 2C) which we can use as a map to predict the effect that different dynamical gradients will have on the observed gene expression patterns.

Analysis of the bifurcations of the system shows a scenario compatible with other TS, in which a continuous variation of AHL or IPTG can change the number of stable states via saddle-node bifurcations (Gardner *et al*, 2000; Guantes & Poyatos, 2008; Perez-Carrasco *et al*, 2016). Starting from a bistable condition, a saddle-node bifurcation occurs through the collision of a stable and an unstable state of the system, leaving only one possible stable state left (Fig 2C). For a cell in the state about to disappear, a small perturbation of IPTG or AHL induces a sharp switch to the opposite state, producing a sharp transition between cell states.

There are two different saddle-node lines in the phase plane (AHL and IPTG) of the bifurcation diagram that collide at a certain

concentration of the inducers (around $10^{-3}$ µM AHL and 10 mM IPTG). Called a cusp bifurcation, this point separates the regions of inducer in the phase plane in which the switch between states occurs through a saddle-node bifurcation (bistability) or through the continuous change of expression of a monostable state. Thus, the availability of a cusp point allows to explore the properties of two different patterning strategies (bistability versus sigmoidal response) for the same circuit topology.

## Controlling the hysteresis, position, and sharpness of the boundary

Next, we analyzed how the IPTG concentration influences the hysteresis of the circuit, characterized by the range of inducer concentrations for which the circuit shows bistability (Fig 2D). Since IPTG is in control of the repression of the green node over the red node, it is a perfect candidate to control the hysteresis of the TS. Based on the bifurcation diagram, we expected that the lower the IPTG concentration (i.e., the stronger the repression on the red node), the bigger the range of bistability. Testing the bistability of the circuit at different IPTG concentrations confirmed this hypothesis, showing a response without bistability to an AHL gradient at high values of IPTG (10 mM) and an increasing range of bistability as IPTG decreases. The highest amount of hysteresis is observed in the absence of IPTG. Here, cells are not able to switch from green to red, even in the presence of high AHL. However, AHL is enough to preserve the red state once reached. We thus observed three different ranges of hysteresis: (i) no hysteresis (> 1 mM IPTG) in a sigmoidal regime; (ii) hysteresis in a bistable regime with the possibility to switch between both states (around 0.125 mM IPTG); and (iii) hysteresis in a bistable regime with irreversibility of the green state (around 0 mM IPTG). Those three regimes were also observed as spatial patterns on the grid, when placed on agar plates with uniform IPTG concentrations (Fig 2D, right).

In addition to controlling the transition between a sigmoidal and a bistable regime and consequently the hysteresis, the IPTG concentration also affects the position of the boundary between the red and green states. Higher concentrations of IPTG allow the

system to switch from green to red at lower AHL concentrations (Fig 2A and D), thus controlling the boundary position for a given AHL gradient. Finally, IPTG also tunes the transition sharpness between the red and green states (Figs 2E and EV4, and Appendix Fig S1). Starting from the green state, at high concentrations of IPTG beyond the cusp point, we observed a sigmoidal expression response in an AHL concentration gradient, similar to the one expected from a circuit with a single repressor (TetR) active. At lower IPTG concentrations, below the cusp point, the system displayed a sharper transition as expected from the saddle-node bifurcation. Moreover, these different transitions are consistent with the distinct population behaviors during cell state transitions along the gradient: a smooth transition of a single population for the sigmoidal behavior and a bimodal population transition (Fig 2A) for the bistable switch behavior. In particular, the bimodal profile is only observed close to the bifurcation, where the timescale of noise-induced transitions between cellular states is smaller than the duration of our observation time (10 h) (Perez-Carrasco et al, 2016). In summary, the regulator IPTG allows us to choose between a bistable and a sigmoidal regime (at high IPTG concentrations) and thus to control the hysteresis, the position, and the sharpness of the boundary.

### Temporal dynamics of the patterning

So far, we investigated the influence of IPTG on the pattern in an AHL gradient by studying the static spatial gene expression after 10-h growth in liquid culture or overnight incubation of the grid. However, in order to understand the process of pattern formation it is of paramount importance to study the temporal dynamics of the patterning process. To this end, we measured the fluorescence of the cells at different concentrations of IPTG and AHL over a time course of 10 h with flow cytometry (Fig 3, Movie EV1, Appendix Fig S2). We observed that gradients inducing a pattern across a bistable region have a slower and position-dependent response than those patterning across a sigmoidal region. In particular, the time to switch from green to red (> 90% red events) depends on the concentration of IPTG, switching at 4 h at high IPTG concentrations (10 to 0.5 mM) and after 6 h and 8 h for lower IPTG concentrations (0.16 and 0.125 mM, respectively; Fig 3A). Similarly, for a constant amount of IPTG, the switching time depends on the AHL concentration, switching at times as slow as 6–10 h for an IPTG concentration corresponding to the bistable region (0.125 mM Fig 3B). On the other hand, when inducing a pattern in the sigmoidal regime (10 mM IPTG), we observe a more consistent switching time (4 h) across different AHL concentrations (Fig 3C).

Interestingly, the behaviors of the switch at the population level over time are identical to the ones observed across different AHL concentrations: At high IPTG (beyond the cusp bifurcation, in the sigmoidal regime), we measured one population moving as a whole, while at lower IPTG concentrations (bistable regime), we observed the cells splitting into two divergent subpopulations (Fig 3A and B, and Movie EV1), suggesting that the dynamics of the sigmoidal patterning are a result of the population relaxing to the unique possible steady state, whereas in the bistable regime, the transition is controlled through the stochastic switching between the two possible states, with a time that is determined by intrinsic noise and

can therefore be slower than the degradation rate of the proteins composing the TS.

### Patterning in the sigmoidal regime is faster than that in the bistable regime

Combining our model with 2D diffusion allowed us to reproduce the patterns observed in the grid assay shown in Fig 1 (Fig 4A). In addition, integrating the diffusion of the inducer with the dynamical properties of the bifurcations of the system can shed light on the different patterning dynamics observed. In particular, consistent with our flow cytometry data (Fig 3), the model suggests that the switching slows down close to the saddle-node bifurcation, a phenomenon called critical slowing down (Narula et al, 2013; Perez-Carrasco et al, 2016). Thus, for different constant values of IPTG, a gradient of AHL is expected to create a moving boundary at different speeds for cells switching from the green to the red state (Fig 4B). To test this prediction, we measured the position of the boundary over time in the grid assay at two different IPTG concentrations corresponding to two different dynamical regimes (sigmoidal, 1 mM and bistable, 0.125 mM; Fig 4B). As expected, we observed that the transition to the production of mCherry starts earlier and advances faster in the sigmoidal regime, equipping the TS with time control of the pattern formation through the cusp bifurcation. This was confirmed through simulation of the diffusion of the inducers on the grid, where the model predicts the same trend than the experimental data, with no more than three squares of difference. On the other hand, in this assay, we did not observe clear differences in the precision of the boundary, potentially due to the high starting cell density.

### Spatiotemporal control of the pattern by using spatially homogenous signals

In addition to the pattern formation through a bistable or monostable sigmoidal regime, the TS offers the possibility to move between both

**Table 1. Parameters**

| Parameter | Median | Credibility interval |
| --- | --- | --- |
| $\alpha_X$ | 364 a.u. | (252, 444) a.u. |
| $\alpha_Y$ | 310 a.u. | (153, 422) a.u. |
| $\tilde{\beta}_X$ | 362 a.u. | (200, 598) a.u. |
| $\tilde{\beta}_Y$ | 438 a.u. | (322, 644) a.u. |
| $K_{IPTG}$ | 2.22 mM$^{-1}$ | (1.04, 3.16) mM$^{-1}$ |
| $K_{AHL}$ | 133 µM$^{-1}$ | (59.5, 318) µM$^{-1}$ |
| $K_{LacI}$ | 4.17 10$^{-2}$ a.u. | (2.11, 8.90) a.u. |
| $K_{TetR}$ | 27.4 10$^{-2}$ a.u. | (6.12, 99.3) 10$^{-2}$ a.u. |
| $n_{LacI}$ | 2.17 | (1.00, 3.73) |
| $n_{AHL}$ | 1.61 | (1.04, 2.23) |
| $n_{TetR}$ | 2.29 | (1.16, 5.01) |

Summary of values of parameters inferred from the experimental data corresponding to the distributions of Fig 3. For each parameter, the median and the 95% credibility interval of each marginal distribution are indicated. The median of each parameter corresponds with the value used in the mathematical model for the rest of simulations of the manuscript.

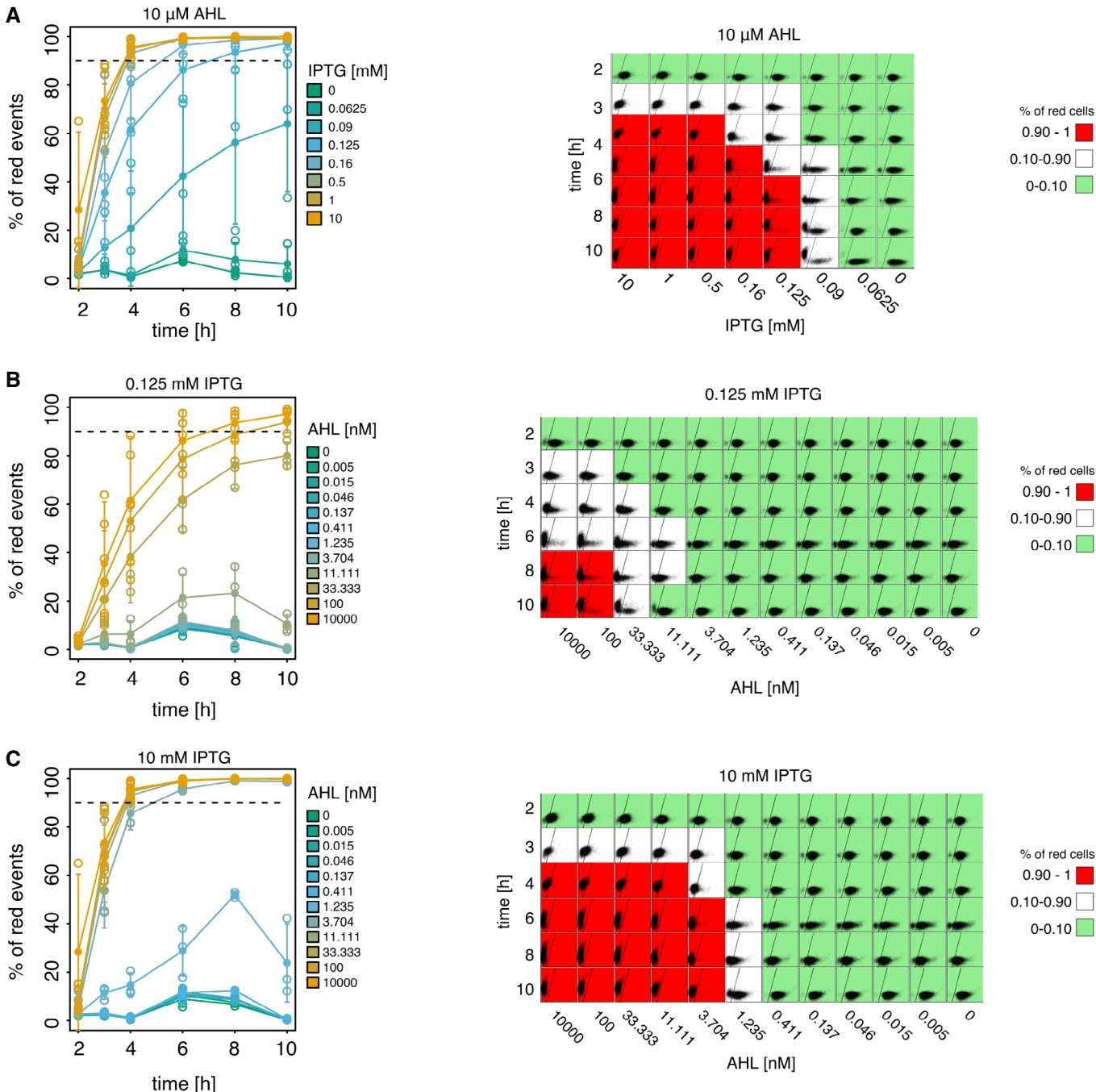

**Figure 3. Temporal dynamics of patterning with the inducible toggle switch.**

A–C  Effect of IPTG and AHL concentrations on the switching time from the green to the red state. For plots in the right column, each square represents flow cytometry data of 10,000 events measuring red (*y*-axis) and green fluorescence (*x*-axis). Background color of each square indicates whether > 90% of the events are in the red or the green gate. Plots in the left column show the percentage of cells in the red gate over time. Shown are the mean (full circles) and standard deviation (error bars) of three biological replicates (individually shown as empty circles). (The data at time point 8 h for 1 and 10 mM IPTG are based on two replicates due to failed flow cytometry measurements.) Dashed lines indicate the 90% threshold used to color the flow cytometry plots. (A) Analysis of state transition over time in the presence of different IPTG concentrations and a high level of AHL (10 μM). (B, C) Analysis of state transition over time in the presence of different AHL concentrations and two different IPTG concentrations corresponding to the bistable (0.125 mM) and sigmoidal (10 mM) regimes.

Source data are available online for this figure.

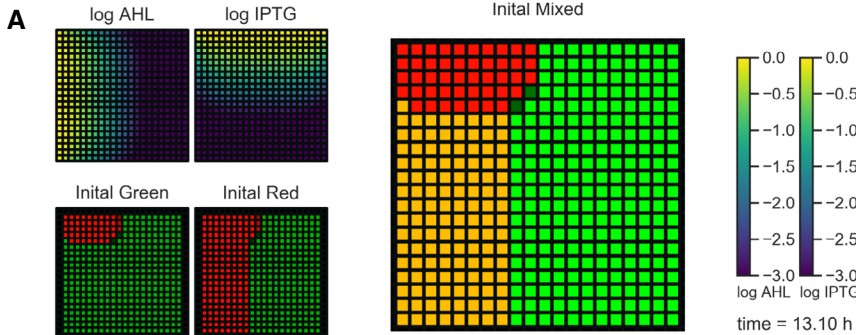

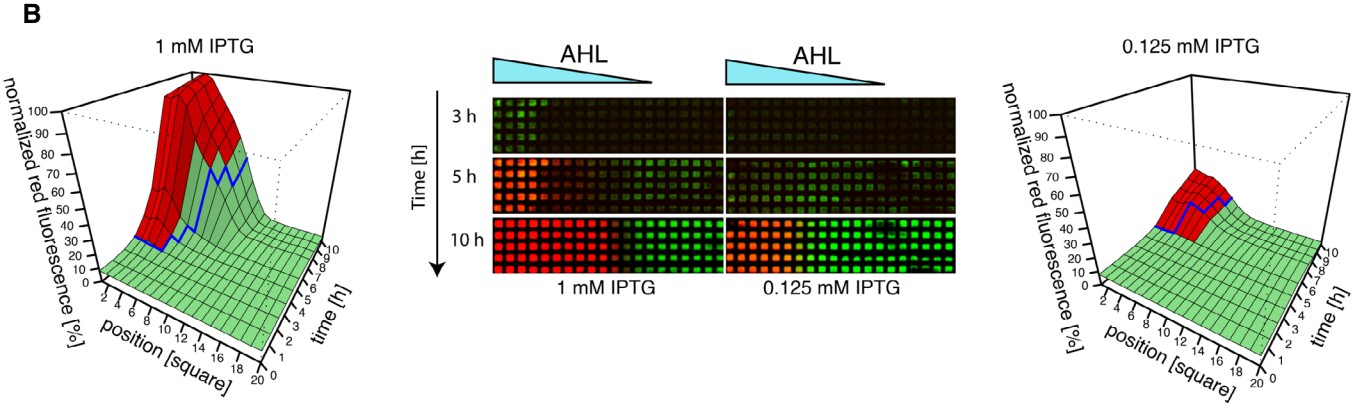

**Figure 4. Patterning in the sigmoidal regime is faster than that in the bistable regime.**

A  Modeling of the pattern observed in the grid assay (Fig 1C–E). Results show the state of the system at 13.10 h posterior to gradient initiation for both diffusive molecules (top left panels) and the response of the network for different initial conditions (bottom left panels and right panel). Gene expression is color-coded shown in green for GFP/LacI expression and in red for mCherry/TetR expression. Positions where different cellular states are obtained depending on the initial condition chosen are indicated in yellow.

B  Time course of pattern formation in the grid in the sigmoidal (1 mM IPTG, left) and bistable (0.125 mM, right) regimes. Center: Before growing on the grid, bacteria were turned into the green state. The indicated concentration of IPTG was homogeneously present in the agar plate, and 5 μl of 100 μM AHL was loaded at the left edge of the grid. The grids were incubated at 37°C and imaged at the indicated times. A representative replicate is shown. Left and right 3D plots represent the quantitative analysis of mean red fluorescence intensity in the grid over time and position of three biological replicates. Blue lines represent the boundary predictions from the mathematical model. Green and red colors correspond to measurements where the red fluorescence intensity is below or above 50% of the highest intensity measured along the AHL gradient at each time point, respectively, corresponding to how the boundary was defined *in silico* (see Materials and Methods for details).

Source data are available online for this figure.

different regimes during the pattern formation, allowing for different patterning strategies that can exploit the properties of both regimes. In particular, by manipulating the homogeneous levels of IPTG in time for a given gradient of AHL, we can control the pattern formation process. An initial population of cells in the green state in the absence of IPTG (bistable irreversible regime) is unaffected by the gradient of AHL. Adding IPTG homogeneously at the desired time point brings the cells to the sigmoidal monostable regime able to respond to the gradient of AHL and forming a boundary. Once the boundary is located at the desired position, removing IPTG takes the system back to the bistable zone, thus freezing the boundary and making it robust to changes in AHL (Fig 5A). Consequently, the system is able to maintain a pattern in the absence of IPTG which removes the requirement of maintaining a precise AHL gradient to keep the pattern boundary. Therefore, a pulse of IPTG can combine advantages of two distinct regimes: of the sigmoidal monostable regime for a fast establishment

of a pattern (Fig 4) and of the irreversible regime to make the pattern robust to changes in the AHL gradient. To demonstrate this effect, we grew cells initially in the green state at different concentrations of AHL and exposed them to pulses (2, 3.5, 5, 6.5 h) of 10 mM IPTG (Fig 5B). We then grew them for further 6 h in the absence of IPTG but kept the AHL concentrations used during the pulse. Indeed, cells receiving enough AHL ($\geq 0.01$ μM) were able to maintain the red state. Interestingly, this was the case even for short pulses where not the whole population did have enough time to switch (2 h). For intermediate values of AHL, close to the boundary, the final state of the population depends on the gene expression of the population with respect to the basins of attraction of the bistable regime. In particular, it is noteworthy that when the single-cell population at the end of the pulse was located at intermediate green and red fluorescence levels, it split into two subpopulations (green or red fluorescence) when brought back to the bistable regime.

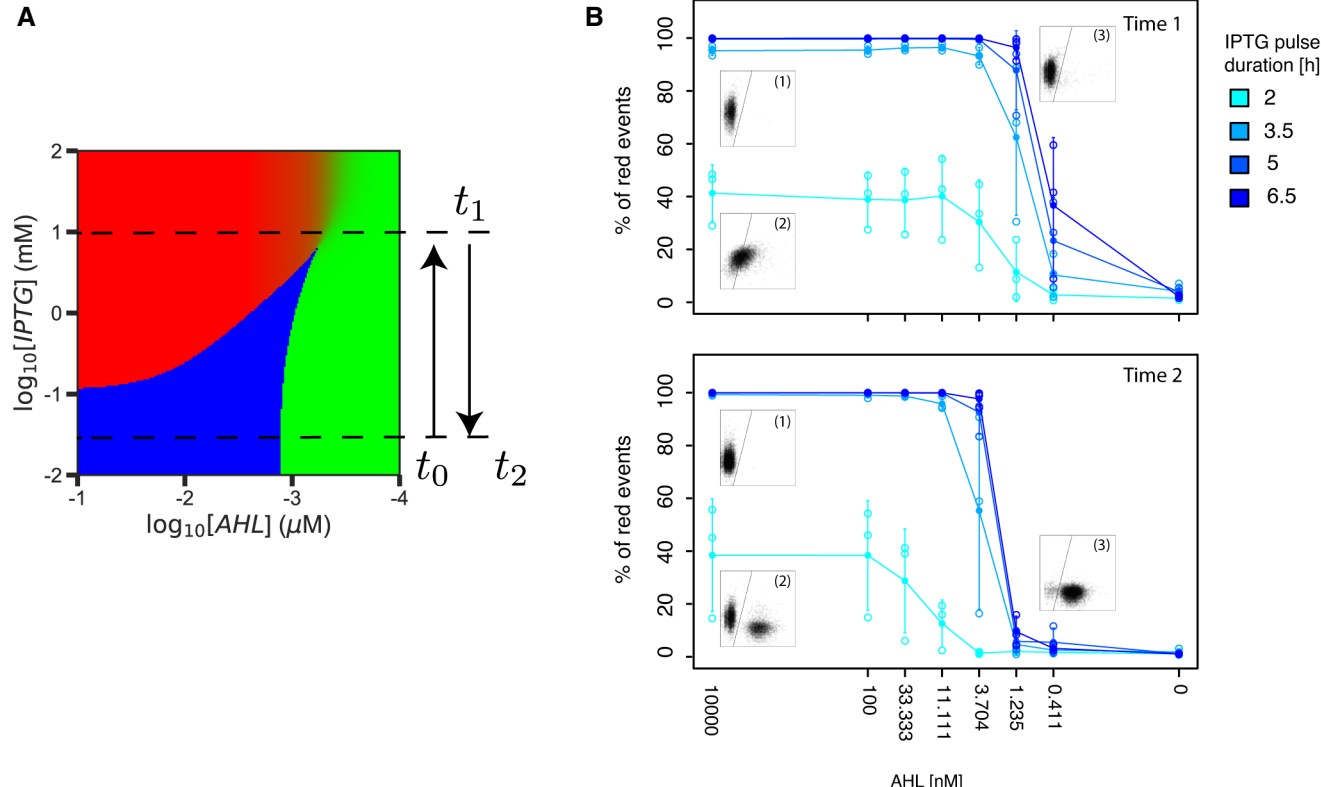

**Figure 5. A homogeneous pulse of IPTG allows to control the pattern formation.**

A  Schematic of the protocol used. Cells initially in the green state for low levels of IPTG (irreversible bistable regime) are exposed to a pulse of high IPTG (at time $t_0$) bringing the system to the sigmoidal region where a gradient of AHL can induce a pattern. Removing IPTG (at time $t_1$) takes the system back to the bistable zone, thus freezing the boundary and making it robust to changes in AHL (at any posterior time $t_2$).

B  Response of the system to IPTG pulses of different durations (2–6.5 h) at different AHL concentrations. Cells were grown in the presence of 10 mM IPTG and the indicated amount of AHL and analyzed by flow cytometry. Shown are the mean (full circles) and standard deviation (error bars) of three biological replicates (individually shown as empty circles). The top graph displays the percentages of red cells observed just after the incubation with IPTG (time 1, $t_1$). Next, the cells were diluted and grown for another 6 h without IPTG, but with the same concentration of AHL as during the IPTG pulse. The bottom graph displays the percentage of red cells observed after this incubation (time 2, $t_2$). Inset graphs represent red (y-axis) and green (x-axis) fluorescence measured by flow cytometry (10,000 events). The following conditions are shown: (i) 6.5 h pulse, 10,000 nM AHL; (ii) 2 h pulse, 10,000 nM AHL; and (iii) 6.5 h pulse, 1.235 nM AHL.

Source data are available online for this figure.

We used the grid assay to further demonstrate this memory property of the TS and to show that the pattern in the bistable regime is indeed robust to changes in the AHL gradient. We patterned a grid as in Fig 1 and transferred the cells onto a new agar plate where the positional information provided by AHL and IPTG was removed (0 mM IPTG, homogeneous concentration of 5 μM AHL). As predicted, the pattern was maintained in the absence of any spatial information (Fig EV5). This result demonstrates that an inducible TS network is capable of interpreting and maintaining spatiotemporal gradients by making use of the memory properties of the circuit.

## Discussion

Systems displaying hysteresis, and in particular the bistable switch, allow for a sharp threshold response that can turn a graded input into a binary output (Sokolowski *et al*, 2012; Perez-Carrasco *et al*, 2016). Interestingly, the mutual repressing motif of the toggle switch

is widely found in natural pattern-forming systems, for example, in networks responsible for patterning the neural tube (Balaskas *et al*, 2012), the dorsal telencephalon (Srinivasan *et al*, 2014), the *Drosophila* embryo segments (Clark, 2017; Verd *et al*, 2019), and the *Xenopus* mesoderm (Saka & Smith, 2007). However, in addition to forming sharp boundaries, the successful formation of a pattern requires the control of the position and timing of the boundary formation. Here, for the first time we characterize a synthetic inducible toggle switch (Fig 1) in order to explore the dynamic patterning properties of the TS gene regulatory network by combining quantitative measurements with a mathematical reconstruction of the underlying dynamical system. Our setup with an AHL gradient controlling the expression level of one of the promoters and an IPTG-tunable repression allowed us to transition between bistable and sigmoidal regimes. This in turn allowed us to show how the hysteresis, position, timing, and precision of the boundary can be controlled, revealing a trade-off between speed and precision of the boundary formation.

We quantified the gene expression at the single-cell level over a wide range of inducer and regulator concentrations and fit the data to parameterize a mathematical model of gene regulation. The resulting bifurcation diagram of the dynamical system provided the mechanistic understanding required to interpret the different spatiotemporal patterns observed and experimental design guidance for pattern formation with the synthetic TS. In particular, this approach allowed us to characterize the mechanism by which the system can transit from a bistable regime to a sigmoidal unimodal response to the inducer, determined by the cusp bifurcation of the dynamical system (Fig 2). Importantly, this was enabled by the choice of focusing on AHL and IPTG gradients. While combining aTc and IPTG gradients would have reduced the asymmetry of the system, we would not have been able to reach the sigmoidal regime (Fig EV1).

From the exploration of the differences in boundary precision of the bistable and sigmoidal regimes, we observed that while the bistable regime provides the sharper boundary (as previously reported (Isalan *et al*, 2005; Lopes *et al*, 2008)), the sigmoidal regime allows for a faster response (Figs 3 and 4). This reveals a trade-off between the timing and precision of the boundary. These observations are in consonance with dynamical system predictions, in which the sharp transition of the bistable regime—through a saddle-node bifurcation—comes at the price of the critical slowing down close to the bifurcation (Narula *et al*, 2013; Perez-Carrasco *et al*, 2016).

In addition, for different levels of a spatially homogeneous regulator, we have been able to control the range of hysteresis and the position of the pattern boundary for a given inducer gradient (Fig 2). This mechanism is analogue to the one proposed by Cohen *et al* (2014) to control the boundary position of patterns regulated by morphogen gradients by changing the affinity of one of the elements to a spatially homogeneous transcription factor. This result highlights the evolutionary potential of the circuit, allowing for kinetic mechanisms of controlling the position of the boundary without compromising other aspects of the boundary or requiring an upstream regulation of the morphogen gradient.

Further to controlling the position of the pattern, the range of bistability can also lead to an irreversible switch for a wide range of the parameter space. Inside this region of the parameter space, any prepattern can be robustly fixed, only requiring the morphogen information for a certain time window (Figs 5 and EV5). Coupled with the dynamics of boundary formation, this provides alternative patterning strategies to the classical static positional information, in which the precision, position, and timing of the boundary can be controlled by the dynamic properties of the upstream signal. This flexibility is of utmost importance in developmental scenarios, where both the morphogen signaling and the patterns are dynamic processes during tissue differentiation, such as the boundary position movement during the gap gene segmentation in *Drosophila* (DiFrisco & Jaeger, 2019) or the adaptation of the signaling of sonic hedgehog morphogen during neural differentiation in the patterning of the neural tube (Balaskas *et al*, 2012). In this latter scenario, the adaptation to the gradient of sonic hedgehog (through Gli signaling) results in a combination of temporal profiles of activation and repression acting on the patterning circuit (Junker *et al*, 2014; Cohen *et al*, 2015). This highlights the importance to understand bistable switches inside a dynamical scenario. While such dynamics are challenging to measure in the developing embryo, our synthetic biology framework allowed for an alternative route toward understanding the dynamics of the TS in pattern formation. However, extending such parallelisms further will require an exhaustive analysis of the different dynamical properties between our synthetic system in *E. coli* and eukaryotic systems. For instance, we expect the approach performed in this manuscript to be valid in eukaryotic systems in which binding kinetics and changes in promoter availability are faster than the patterning time, but not in situations where they are slower (Bintu *et al*, 2016). Differences might also come from other characteristics of the GRN, such as chromatin remodeling, binding dynamics, or transcriptional bursting (Oates, 2011; Bentovim *et al*, 2017; Mathur *et al*, 2017; Folguera-Blasco *et al*, 2019).

Our improved understanding of the TS for pattern formation opens up new avenues of research. In particular, for many bistable parameter conditions, single-cell expression showed the coexistence of two subpopulations of cells at each one of the available stable states. This provides evidence of the relevance of intrinsic noise in the establishment of the pattern. While previous *in silico* research shows that intrinsic noise can determine the position and precision of morphogen-driven boundaries (Weber & Buceta, 2013; Perez-Carrasco *et al*, 2016), the actual role of noise in the dynamics of pattern formation in living systems and the possibility of optimal dynamical strategies based on the stochasticity of gene expression remain still a conundrum. In addition, it poses new challenges to dynamical system inference in which different sources of intrinsic noise must be disentangled from measurement noise in order to obtain an accurate characterization of the circuit (Dharmarajan *et al*, 2019).

Overall, our results underscore the relevance of studying the dynamical context of a gene regulatory network in order to understand patterning processes, not only of synthetic circuits but also of developmental systems (Ferrell, 2002; Sagner & Briscoe, 2017). Future work will reveal if the TS properties are conserved when incorporated in more complex (synthetic) gene regulatory networks, for example, when combined with the repressilator (Elowitz & Leibler, 2000; Potvin-Trottier *et al*, 2016; preprint: Santos-Moreno & Schaerli, 2019a) to yield the AC-DC network (Balaskas *et al*, 2012; Panovska-Griffiths *et al*, 2013; Perez-Carrasco *et al*, 2018; Verd *et al*, 2019). Moreover, the here established engineering guidelines on how to control patterning with a synthetic TS will be valuable for future synthetic pattern formation, for example, in the context of engineered living materials based on bacterial biofilms (Cao *et al*, 2017; Gilbert & Ellis, 2018; Nguyen *et al*, 2018; Moser *et al*, 2019).

# Materials and Methods

### Media

Cloning experiments used lysogeny broth medium (LB: 10 g Bacto-Tryptone, 5 g yeast extract, 10 g NaCl per 1 l) supplemented with the appropriate antibiotic (25 μg/ml kanamycin or 25 μg/ml spectinomycin). All experiments with the synthetic circuit were performed in M9 minimal medium (1× M9 salts, 2 mM MgSO$_4$, 0.2 mM CaCl$_2$, 0.0005% (w/v) thiamine) with 0.2% (w/v) glucose as carbon source, supplemented with 0.1% (w/v) casamino acids and the appropriate antibiotics (25 μg/ml kanamycin and 25 μg/ml spectinomycin).

### Reagents

Restriction enzymes, alkaline phosphatase from calf intestine (CIP), DNA polymerase I large (Klenow) fragment, and T4 DNA ligase were purchased from New England Biolabs (NEB). Oligonucleotides and chemicals were obtained from Sigma-Aldrich. Polymerase chain reactions (PCRs) were carried out with Q5 Hot Start High-Fidelity DNA Polymerase (NEB). Colony PCRs were performed with Taq polymerase (NEB). PCR products and digested plasmids were purified with the Monarch PCR & DNA Cleanup Kit (NEB). Plasmids were purified using the QIAprep Spin Miniprep Kit (QIAGEN).

### Cloning of the inducible toggle switch circuit

We cloned the morphogen-inducible toggle switch plasmid ("TS_pLuxLac") from an already functional toggle switch plasmid (pKDL071) (Litcofsky *et al*, 2012), kindly provided by Jeong Wook Lee. The two Ptrc-2 promoters upstream of TetR and mCherry coding sequences were replaced by the hybrid promoter pLuxLac (BBa_I751502) with the plug-and-play method initially used to assemble pKDL071 using the restrictions sites NcoI and SalI upstream of TetR and XmaI and MfeI upstream of mCherry.

For the "pCDF_luxR" plasmid, the pLuxL promoter (BBa_R0063) and the LuxR gene (synthesized by GenScript) were introduced into a pCDF plasmid with a customized multiple cloning site (Schaerli *et al*, 2014) between the KpnI and BamHI sites. The sequences and plasmids are available through Addgene, ID 140426 for TS_pluxlac and ID 140039 for pCDF_luxR.

### Strain and growth condition

pCDF_luxR and TS_pLuxLac were transformed into the *E. coli* strain MK01 (Kogenaru & Tans, 2014). The absence of the *lacI* gene in this strain avoids unexpected cross talk between the synthetic circuit and the host.

Bacteria were turned into red state by inoculating single colonies into 4 ml M9 medium in the presence of 1 mM IPTG and 10 μM AHL (*N*-(β-Ketocaproyl)-L-homoserine lactone). They were grown overnight at 37°C and 200 rpm shaking. The same procedure was used to turn the bacteria into the green state, with the difference that the medium did not contain IPTG and AHL. These bacteria were plated out on LB agar plates (supplemented with 10 μM AHL for the red state) and incubated overnight at 37°C. The plates were stored at 4°C, and single colonies were picked for the experiments.

### Flow cytometry

For each biological replicate, a single colony in the red or green state was cultured in M9 for 4–6 h and put at 4°C before entering stationary phase (below 0.8 OD). The following day, these cultures were diluted to 0.01 OD (NanoDrop 2000, Thermo Fisher) and added into the wells of a 96-well plate (CytoOne) to a total volume of 120 μl including indicated concentrations of AHL and IPTG. The plate was incubated in a BioTek Synergy H1 Plate Reader at 37°C with 548 cpm (2 mm) double orbital shaking speed. Absorbance (600 nm) was monitored every 10 min to check that cells did not enter stationary phase (below 0.3 in the plate reader) as cells in stationary phase can no longer switch between the two states. After

5 h, cells were diluted 100 times before incubating them again under the same conditions for a total of 10 h.

At indicated times, 5 μl of the cell cultures was diluted into 95 μl of phosphate-buffered saline (PBS) and analyzed by flow cytometry (BD LSRFortessa™) with 488-nm excitation and FITC filter for measuring GFP fluorescence and 561-nm excitation and PE-Texas Red filter for measuring mCherry fluorescence. Ten thousand events were recorded and analyzed by FlowJo and a custom-made R script.

First, cells were gated with FlowJo in the SSC-H and FSC-H scatter plot. Next, FITC-H and PE-Texas Red-H data were exported to be analyzed in R. The red and green gates were set with the help of positive controls for red and green fluorescence, so that nearly 100% of the events lay in the respective gate. These controls were an overnight culture of our bacteria in the presence of 1 mM IPTG and 10 μM AHL for red and without any inducer for green. In the figures, percentages of red cells correspond to the percentages of cells found in the red gate and the red fluorescence mean corresponds to the mean of red fluorescence of all the gated cells normalized to the red fluorescence of the cells grown at the highest AHL and IPTG concentrations and corrected for the background fluorescence (minimal red fluorescence value measured in the experiment).

For the IPTG time pulse experiment, we started the experiment with a culture of OD 0.1 in a medium containing 10 mM IPTG and the indicated concentration of AHL. The samples were incubated as described above until the first time point (2 h), and an aliquot was stored at 4°C. Then, the cells were diluted 1:5 into fresh medium containing the same IPTG and AHL concentrations and further incubated until the next time point (3.5 h). This procedure was repeated for all time points. An aliquot of each sample was analyzed by flow cytometry once all samples were collected. The next day, the cells were diluted 1,000 times into fresh medium with the indicated concentration of AHL and no IPTG. The cells were grown for 6 h and directly measured by flow cytometry.

### Grid assay

Single colonies in the red or green state were cultured in M9 for 4–6 h and put at 4°C before entering stationary phase (below 0.8 OD). The following day, these cultures were washed from IPTG and AHL by centrifugation for 1 min at 13,000 RPM and resuspended in fresh medium without IPTG or AHL. Then, cells were diluted to an OD of 0.1 before being pipetted onto the grid (ISO-GRID from NEOGEN) (Grant *et al*, 2016) (20 μl of cells was loaded for 10 lines, approximately 0.05 μl per square) that was placed on a M9 agar plate with 0.2% (w/v) glucose and appropriate antibiotics (25 μg/ml kanamycin and 25 μg/ml spectinomycin). The inducers (100 mM IPTG and 100 μM AHL, 5 μl each) were added as indicated in the figures. The plate was incubated overnight at 37°C before green and red fluorescence were measured with a Fusion FX (VILBER) imaging system. We used 0.2-ms exposure with blue light (480 nm) and a F-565 filter for the GFP measurement and 1-min exposure with red light (530 nm) and F-740 filter for mCherry. ImageJ software (R Core Team, 2017) was used to analyze the picture.

For the time course experiment, bacterial solutions with an OD of 3 were loaded on the grid in order to be able to quantify early time points. Fluorescence intensity values for each square of the grids were extracted with a custom Fiji ImageJ (Schindelin *et al*,

2012) macro script, and the maximum value for each square was normalized to the highest measured fluorescence in all the conditions and replicates. The data were plotted with R software (R Core Team, 2017).

## Model derivation and parameterization

We modeled the expression dynamics of the TS by describing the change in time of the concentration of the [LacI] and [TetR] as a balance between their regulated production and degradation. In addition, we made the assumptions that the dynamics of the transcripts and promoter binding/unbinding is faster than the dynamics of production and degradation of the repressor proteins and that the reporters follow the same dynamics as their associated repressors.

The expression of LacI is regulated by TetR, which can inhibit the production of LacI by binding to the TetO promoter. Modeling this interaction as a repressive Hill function we described the evolution in time of LacI as,

$$\frac{d[\text{LacI}]}{dt} = \frac{\beta_Y}{1 + (K_{\text{TetR}}[\text{TetR}])^{n_{\text{TetR}}}} - \delta_Y[\text{LacI}], \tag{1}$$

where $\beta_Y$ is the maximum production of LacI in the absence of the repressor TetR, $K_{\text{TetR}}$ sets the TetR concentration required to halve the production rate of LacI, $n_{\text{TetR}}$ is the Hill coefficient, and $\delta_Y$ is its degradation rate.

On the other hand, the expression of TetR is regulated at the same time by the repression of free LacI in the system [LacI$_f$] and the activation by AHL through the LuxR-AHL complex. Since the presence of free [LacI$_f$] is enough to silence the expression of [TetR] even in the presence of AHL, the production can be modeled as the product of two Hill functions with affinities $K_{\text{LacI}}$ and $K_{\text{AHL}}$, and Hill coefficients $n_{\text{LacI}}$ and $n_{\text{AHL}}$,

$$\frac{d[\text{TetR}]}{dt} = \frac{\beta_X}{1 + (K_{\text{LacI}}[\text{LacI}_f])^{n_{\text{LacI}}}} \frac{(K_{\text{AHL}}[\text{AHL}])^{n_{\text{AHL}}}}{1 + (K_{\text{AHL}}[\text{AHL}])^{n_{\text{AHL}}}} - \delta_X[\text{TetR}], \tag{2}$$

where $\beta_X$ is the production rate in the absence of LacI and saturating amounts of AHL, and $\delta_X$ is the degradation rate of TetR. The amount of free LacI ([LacI$_f$]) is controlled by IPTG, which can bind free LacI with an equilibrium constant $K_{\text{IPTG}}$ impeding the binding with the LuxLac promoter,

$$[\text{LacI}_f] = \frac{[\text{LacI}]}{1 + K_{\text{IPTG}}[\text{IPTG}]}. \tag{3}$$

In order to parameterize the mathematical model equations (1–3), we compared the experimentally observed cellular states with the stable steady states of the theoretical dynamical system for different sets of concentrations of AHL and IPTG. The experimental steady states were obtained by using the gated expression in cellular populations (see Flow Cytometry for details). For each gated population, the state was accepted when it contained at least 15% of the cellular population.

The stable states for the mathematical model ([LacI]*,[TetR]*) were computed by solving equations (1–3) at equilibrium condition

d[LacI]/dt = d[TetR]/dt = 0,

$$[\text{LacI}]^* = f_1([\text{TetR}]^*) = \frac{\beta_Y/\delta_Y}{1 + (K_{\text{TetR}}[\text{TetR}]^*)^{n_{\text{TetR}}}},$$

$$[\text{TetR}]^* = f_2([\text{LacI}]^*; [\text{IPTG}], [\text{AHL}])$$

$$= \frac{\beta_X/\delta_X}{1 + \left(\frac{K_{\text{LacI}}[\text{LacI}]^*}{1 + K_{\text{IPTG}}[\text{IPTG}]}\right)^{n_{\text{LacI}}}} \frac{(K_{\text{AHL}}[\text{AHL}])^{n_{\text{AHL}}}}{1 + (K_{\text{AHL}}[\text{AHL}]^{n_{\text{AHL}}})}.$$

Thus, finding the available steady states for a given condition is reduced to finding the roots of $G(x)$ in the 1-dimensional equation $G([\text{LacI}]^*) = f_1 (f_2 ([\text{LacI}]^*;[\text{IPTG}], [\text{AHL}])) - [\text{LacI}]^* = 0$. This was done by finding the number and approximate location of the roots by evaluating the sign of $G(x)$ over a logarithmically spaced set along the possible values of $[\text{LacI}]^* = [f_1 (\beta_X/\delta_X), \beta_Y/\delta_Y]$. All the values found were refined by using the Brent–Dekker method with hyperbolic extrapolation. Finally, the stability of all the possible found states was addressed by evaluating the eigenvalues of the Jacobian corresponding to equations (1 and 2).

In order to compare the computational steady states with fluorescence measurements, we assumed a linear relationship between fluorescence and gene expression,

$$F_{\text{RED}} = \alpha_Y + \omega_Y[\text{LacI}]^*, \quad F_{\text{GREEN}} = \alpha_X + \omega_X[\text{TetR}]^*.$$

Thus, the parameterization of the problem is reduced to the inference of 11 irreducible identifiable set of parameters $\theta = \alpha_X, \alpha_Y, \tilde{\beta}_X, \tilde{\beta}_Y, K_{\text{TetR}}, K_{\text{LacI}}, K_{\text{AHL}}, K_{\text{IPTG}}, n_{\text{TetR}}, n_{\text{LacI}}, n_{\text{AHL}}$, where $\tilde{\beta}_X$ and $\tilde{\beta}_Y$ are the non-dimensionalizing production rates summarizing the parameter products $\tilde{\beta}_X = \omega_X \beta_X/\delta_X$ and $\tilde{\beta}_Y = \omega_Y \beta_Y/\delta_Y$.

The ensemble of parameters in the mathematical model compatible with experimental observations was inferred using Markov chain Monte Carlo. In particular, we made use of multiple-try differential evolution adaptive Metropolis algorithm (Laloy & Vrugt, 2012) using the PyDREAM implementation (Shockley et al, 2018). The log-likelihood function used to evaluate the goodness of a given set of parameters is defined as,

$$\mathcal{L}(\theta|\text{data}) = \sum_{i=1}^{N} \left( p_i + \min_j \left[ \left( \frac{F_{\text{RED}_{\text{exp}}}^i - F_{\text{RED}_{\text{theo}}}^{i,j}}{\sigma_{\text{RED}}^i} \right)^2 + \left( \frac{F_{\text{GREEN}_{\text{exp}}}^i - F_{\text{GREEN}_{\text{theo}}}^{i,j}}{\sigma_{\text{GREEN}}^i} \right)^2 \right]^2 \right)$$

where the index $i$ in the sum runs over all the experimentally detected cell states for each experimental condition. The experimental fluorescence $F_{\text{RED}_{\text{exp}}}^i, F_{\text{GREEN}_{\text{exp}}}^i$ corresponds with the median of the gated population of cells for the $i$-th observed state. Similarly, $\sigma_{\text{RED}_{\text{exp}}}^i$ and $\sigma_{\text{GREEN}_{\text{exp}}}^i$ are the standard deviation of the gated populations of each state. The theoretical prediction for each parameter set $\theta$ and conditions of state $i$ is given by $F_{\text{RED}_{\text{theo}}}^{i,j}$ and $F_{\text{GREEN}_{\text{theo}}}^{i,j}$, where the superindex $j$ presents the different theoretically predicted stable states in the case of bistability. Finally, parameter $p_i$ penalizes that for the condition given by state $i$ (concentrations of AHL and IPTG), the number of different states detected experimentally is not the same as the number of stable states predicted in the mathematical

model. Thus, $p_i$ promotes parameter sets that match monostable and bistable regions between experimental and computational steady states. In particular, $p_i = 0$ if the number of stable steady states matches, and $p_i < 0$ otherwise. For the inference used in the manuscript, a value of $p_i = -10^2$ was used. The inferred parameters are summarized in Table 1.

**Morphogen diffusion**

The diffusion of the morphogens (AHL, IPTG, and aTc) on the agar plate was assumed to be a 2-D free diffusion determining the concentration of each chemical $\rho(x, y, t)$ at different positions of the plate. This was modeled through the partial differential equations

$$\frac{\partial \rho_{\text{AHL}}}{\partial t} = D\left(\frac{\partial^2 \rho_{\text{AHL}}}{\partial x^2} + \frac{\partial^2 \rho_{\text{AHL}}}{\partial y^2}\right),$$

$$\frac{\partial \rho_{\text{IPTG}}}{\partial t} = D\left(\frac{\partial^2 \rho_{\text{IPTG}}}{\partial x^2} + \frac{\partial^2 \rho_{\text{IPTG}}}{\partial y^2}\right).$$

This equation was solved by discretizing the space using the square experimental grid. The Dirichlet boundary conditions imposed by the experiment are given by the constant concentration of morphogen along one side of the square $\rho(0, y, t) = \rho_0$, and assuming sinks in the rest of the perimeter of the square $\rho(L, y, t) = \rho(x, 0, t) = \rho(x, L, t) = 0$, where $L$ is the length of the side of the grid. In order to take into account the dilution effect of the initial concentration pipetted $c_0$ on the agar before establishing the gradient, we set $\rho_0 = c_0\xi$. Parameters used for the grid assays are $\xi = 10^{-2}$ and $D = 2.5 \ 10^{-3}$ cm$^2$/h (Basu *et al*, 2005; Miyamoto *et al*, 2018). Finally, in order to provide a timescale to the dynamics of protein, not available from the MCMC fitting of the steady states of the system, the degradation rate of the LacI and TetR was set to $\delta = 8.3$ h$^{-1}$ (Wu *et al*, 2011).

Boundary position in the grid model was computed by analyzing the red fluorescence levels along the central strip of the grid TeTR $(x, L/2, t)$. For each time point, a boundary was considered when the difference in fluorescence at both sides of the gradient was above a certain threshold $|$TeTR$(\ell/2, L/2, t) - $TeTR$(L - \ell/2, L/2, t)| > 250$, where $\ell$ is the distance between two adjacent cells of the grid. The position of the boundary $x_b$ was set as the last cell of the grid where the concentration is above the mean point of the fluorescence range for each given time point (TeTR$(\ell/2, L/2, t) + $TeTR$(L - \ell/2, L/2, t))/2$.

## Data availability

The plasmids and their full sequences are available via Addgene (ID 140426 and ID 140039). The source data for all figures are provided as Supplementary documents. The raw flow cytometry data underlying Fig 2 and 3 are provided as Dataset EV1. The raw flow cytometry data underlying Fig 5 are provided as Dataset EV2. The computer code used to simulate the dynamical system and infer the model parameters can be found at: https://github.com/2piruben/SyntheticBistableSwitch.

**Expanded View** for this article is available online.

## Acknowledgements

We thank Florence Gauye for excellent technical assistance and all Schaerli laboratory members for useful discussions. This work was funded by Swiss National Science Foundation Grant 31003A_175608 and an IPhD SystemsX.ch grant. RP-C acknowledges UCL Mathematics Clifford Fellowship.

## Author contributions

IB, RPC, and YS designed research. IB performed experiments and analyzed data. RPC carried out the mathematical modeling and the data fitting. YS supervised the project. IB, RPC, and YS wrote the manuscript. All authors have given approval to the final version of the manuscript.

## Conflict of interest

The authors declare that they have no conflict of interest.

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
