## [Review Process File · Molecular Systems Biology]

Controlling spatiotemporal pattern formation in a concentration gradient with a synthetic toggle switch

Içvara Barbier, Ruben Perez-Carrasco, and Yolanda Schaerli

DOI: [10.15252/msb.20199361](https://doi.org/10.15252/msb.20199361)

Corresponding author(s): Yolanda Schaerli (yolanda.schaerli@unil.ch) , Ruben Perez-Carrasco (r.carrasco@ucl.ac.uk)

Review Timeline:

Submission Date:	14th Nov 19
Editorial Decision:	19th Jan 20
Revision Received:	13th Mar 20
Editorial Decision:	25th Apr 20
Revision Received:	6th May 20
Accepted:	8th May 20

Editor: Jingyi Hou

Transaction Report:

Thank you for submitting your work to Molecular Systems Biology. We have now heard back from two of the three reviewers who agreed to evaluate your manuscript. Unfortunately, after a series of reminders we did not manage to obtain a report from reviewer #1. In the interest of time, and since their recommendations are quite similar, I prefer to make a decision now rather than further delaying the process. If we receive the comments from reviewer #1, we will send them to you, and you can address the issues raised by reviewer #1 together with those raised by the other two reviewers. You will see from the comments below that reviewer #2 and #3 find the manuscript to be of interest. They raise, however, several important points, which should be convincingly addressed in a revision of this work.

I think that the reviewers' recommendations are rather clear and there is no need to reiterate their comments. Importantly, considering both reviewers pointed out that overall novelty is modest, the concerns raised by reviewer #2 regarding the implications of the presented findings in understanding pattern formation in a biological context (e.g. animal development etc.) need to be addressed, in order to enhance the conceptual novelty and the level of biological insight provided by the study.

All other issues raised by the reviewers need to be satisfactorily addressed as well. As you may already know, our editorial policy allows in principle a single round of major revision and it is therefore essential to provide responses to the reviewers' comments that are as complete as possible. Please feel free to contact me in case you would like to discuss in further detail any of the issues raised by the reviewers.

On a more editorial level, we would ask you to address the following issues:

- Please provide a .docx formatted version of the manuscript text (including legends for main figures, EV figures and tables). Please make sure that the changes are highlighted to be clearly visible.
- Please provide individual production quality figure files as .eps, .tif, .jpg (one file per figure).
- Please provide a .docx formatted letter INCLUDING the reviewers' reports and your detailed point-by-point responses to their comments. As part of the EMBO Press transparent editorial process, the point-by-point response is part of the Review Process File (RPF), which will be published alongside your paper.

-Please note that all corresponding authors are required to supply an ORCID ID for their name upon submission of a revised manuscript.

-We have replaced Supplementary Information by the Expanded View (EV format). In this case, since you may provide more figures during revision, all of them can be included in a PDF now called Appendix. Appendix figures (and Tables) should be labeled and called out as: "Appendix Figure S1, Appendix Figure S2... Appendix Table S1..." etc. Each legend should be below the corresponding Figure/Table in the Appendix. Please include a Table of Contents in the beginning of the Appendix. For detailed instructions regarding expanded view please refer to our Author Guidelines: (<https://www.embopress.org/page/journal/17444292/authorguide#expandedview>).

-- Before submitting your revision, primary datasets and computer code produced in this study need to be deposited in an appropriate public database (see <https://www.embopress.org/page/journal/17444292/authorguide#dataavailability>). - Dataset #1
- Dataset #2>

The accession numbers and database should be listed in a formal "Data Availability" section (placed after Materials & Method) that follows the model below (see also <https://www.embopress.org/page/journal/17444292/authorguide#dataavailability>). Please note that the Data Availability Section is restricted to new primary data that are part of this study.

Data availability

- We would encourage you to include the source data for figure panels that show essential quantitative information. Additional information on source data and instruction on how to label the files are available at < <https://www.embopress.org/page/journal/17444292/authorguide#sourcedata> >.

- All Materials and Methods need to be described in the main text. We would encourage you to use 'Structured Methods', our new Materials and Methods format. According to this format, the Material and Methods section should include a Reagents and Tools Table (listing key reagents, experimental models, software and relevant equipment and including their sources and relevant identifiers) followed by a Methods and Protocols section in which we encourage the authors to describe their methods using a step-by-step protocol format with bullet points, to facilitate the adoption of the methodologies across labs. More information on how to adhere to this format as well as downloadable templates (.doc or .xls) for the Reagents and Tools Table can be found in our author guidelines: < <https://www.embopress.org/page/journal/17444292/authorguide#researcharticleguide> >. An

example of a Method paper with Structured Methods can be found here: .

- Please provide a "standfirst text" summarizing the study in one or two sentences (approximately 250 characters, including space), three to four "bullet points" highlighting the main findings and a "synopsis image" (550px width and max 400px height, jpeg format) to highlight the paper on our homepage.

-When you resubmit your manuscript, please download our CHECKLIST (http://embopress.org/sites/default/files/Resources/EP_Author_Checklist.xls) and include the completed form in your submission. *Please note* that the Author Checklist will be published alongside the paper as part of the transparent process <http://msb.embopress.org/authorguide#transparentprocess>.

If you feel you can satisfactorily deal with these points and those listed by the referees, you may wish to submit a revised version of your manuscript. Please attach a covering letter giving details of the way in which you have handled each of the points raised by the referees. A revised manuscript will be once again subject to review and you probably understand that we can give you no guarantee at this stage that the eventual outcome will be favorable.

My apologies once again for the delay.

Yours sincerely,

Jingyi Hou
Editor
Molecular Systems Biology

If you do choose to resubmit, please click on the link below to submit the revision online *within 90 days*.

Link Not Available

IMPORTANT: When you send your revision, we will require the following items:

1. the manuscript text in LaTeX, RTF or MS Word format
2. a letter with a detailed description of the changes made in response to the referees. Please specify clearly the exact places in the text (pages and paragraphs) where each change has been made in response to each specific comment given
3. three to four 'bullet points' highlighting the main findings of your study
4. a short 'blurb' text summarizing in two sentences the study (max. 250 characters)
5. a 'thumbnail image' (550px width and max 400px height, Illustrator, PowerPoint or jpeg format), which can be used as 'visual title' for the synopsis section of your paper.
6. Please include an author contributions statement after the Acknowledgements section (see <https://www.embopress.org/page/journal/17444292/authorguide>)
7. Please complete the CHECKLIST available at (<http://bit.ly/EMBOPressAuthorChecklist>). Please note that the Author Checklist will be published alongside the paper as part of the transparent process

(<https://www.embopress.org/page/journal/17444292/authorguide#transparentprocess>).

8. Please note that corresponding authors are required to supply an ORCID ID for their name upon submission of a revised manuscript (EMBO Press signed a joint statement to encourage ORCID adoption). (<https://www.embopress.org/page/journal/17444292/authorguide#editorialprocess>)

Currently, our records indicate that the ORCID for your account is 0000-0002-9083-7343.

Link Not Available

The system will prompt you to fill in your funding and payment information. This will allow Wiley to send you a quote for the article processing charge (APC) in case of acceptance. This quote takes into account any reduction or fee waivers that you may be eligible for. Authors do not need to pay any fees before their manuscript is accepted and transferred to the publisher.

*** PLEASE NOTE *** As part of the EMBO Press transparent editorial process initiative (see our Editorial at <http://dx.doi.org/10.1038/msb.2010.72>), Molecular Systems Biology publishes online a Review Process File with each accepted manuscripts. This file will be published in conjunction with your paper and will include the anonymous referee reports, your point-by-point response and all pertinent correspondence relating to the manuscript. If you do NOT want this File to be published, please inform the editorial office at msb@embo.org within 14 days upon receipt of the present letter.

Reviewer #2:

In the manuscript "Controlling spatiotemporal pattern formation in a concentration gradient with a synthetic toggle switch," the author's explore how a "classical" regulatory motif, the toggle switch, composed of two cross-repressing transcription factors, may convert continuous information provided by the gradient into discrete stripes of gene expression. Towards this end, the authors build a synthetic biology framework to understand and characterize the spatiotemporal patterning properties of the toggle switch. They build a synthetic toggle switch controllable by diffusible molecules in *Escherichia coli* and explore the "patterning" by combining quantitative measurements with a mathematical reconstruction of the system. The authors find that the toggle switch can produce robust patterns with sharp boundaries, bistability, and hysteresis in a tunable manner.

A key conclusion is that by building a mathematical representation-namely bifurcation diagrams of the system-they were able to explore the various spatial-temporal patterns observed and experimental "guide" pattern formation. A key finding was that they could characterize the switch from bistability to sigmoidal response. In sum, the methodology combining modeling with their grid-based systems and flow-cytometry was excellent and allowed the exploration of a classic network motif.

While the work is compelling, and the conclusions appear solid, the main shortcoming is that many aspects of this work are not entirely novel and may not represent a significant advance in the field.

For example, a bistable genetic switch based on designable DNA-binding domains has been explored by Lebar and colleagues (Nat. Comm. 2014). Similar to the present work, these authors found non-linearity that results in epigenetic bistability. Others have also built bistable systems that not switchable, depending on the initial input conditions (Kim et al., 2006). See also Pokhilko et al., 2019.

Finally, while the descriptions of developmental systems are interesting, the authors should highlight the caveats of their system. Namely, the transcription within animal systems is highly dynamic, with rapid binding kinetics. For example, Sox2 dwells at specific target DNA for only ~12.0~14.6s. Could their system capture this? If yes, they should describe it.

While the authors did not observe a clear difference in the precision of the boundary, it would be great to explore this further. Could they be more precise in the concentration gradients? Animal development is often not precise on the input side so could this be a regulatory mechanism?

In sum, this manuscript provides a fantastic ability to control the transition between regime modes (sigmoidal, bistable, and hysteresis). However, the authors would need to justify the conceptual advances in the field. Simply calling varying IPTG conditions a "morphogen" is not enough to justify how this differs from prior work.

Reviewer #3:

This paper elegantly combines experiments with mathematical modeling to study in a systematic manner the behavior of a synthetic toggle switch implemented in the bacterium *E. coli*. Specifically, the authors study the response of this circuit to spatial gradients of several inducers, which they interpret as morphogens. From this analysis, they extract conclusions relative to the stripe-forming ability of the circuit in response to those gradients in different parameter regimes, regarding aspects such as its robustness, controllability, and kinetics. While the circuit employed is rather standard, and the system is quite far from realistic situations displaying morphogen-driven pattern formation (in development, for instance), the systematic study presented here has value in itself, constituting almost a textbook analysis (which in fact I'll probably use in my classes in the future, given that the preprint is already publicly available in bioRxiv). There are some issues, however, that make the paper somewhat unclear, and which I would ask the authors to address before publication:

1.- According to the text, the inducers are added instantaneously on the corner of the hydrophobic grid, and then the system is left to relax for hours. If that is the case, the gradients are continuously decaying, and the time at which the experimental observations of Figs. 1, 2 and S1-S4 is made should play an important role. The authors state that the measurements are taken after overnight incubation, but this is too vague in my opinion. What's the exact time? What happens if that time is doubled, for instance? And why can these results be fitted to the steady state of their model? These questions should be substantially clarified in the manuscript.

2.- How is the boundary for the gating of the flow cytometry measurements chosen? It would be helpful if this were also clarified in the text.

3.- One point that I find interesting is the difference between bistability and bimodality reported in Fig. 2A. I guess that the reason for this difference is the fact that the potential barrier between the

two stable states in part of the bistable regime is too large for jumps to take place by the time the measurements are taken. In fact the authors invoke this when comparing between the sigmoidal and bistable regimes, but I would expect that even within the bistable regime itself one could find the two behaviors, depending on the time at which the measurements are made. In fact, I would expect the two regions (bimodality and bistability) to grow more similar to each other the later the measurements are made (with a limit given by the time at which the gradients virtually disappear due to diffusion). Am I correct? If so, I would ask the authors to clarify this in the text. On the other hand, if I'm wrong and the observations made here can be interpreted in terms of a steady state behavior, it would be nice to calculate the effective potential of the system as was previously done by Wang et al in 2010 (DOI 10.1016/j.bpj.2010.03.058).

4.- This system has three different inductors, namely AHL, IPTG and aTc. Given the symmetry of the mutual inhibition switch, the natural two parameters to focus on would be, in my opinion, IPTG and aTc, whereas AHL would be a general inducer of the circuit. However, the authors decided to focus mainly on AHL and IPTG, and this asymmetry leads to the asymmetry reported on page 6: "The transition from red to green is less affected by the concentrations of IPTG than the transition from green to red, because of the asymmetry of the network". Can the authors confirm whether the need to focus on initial green states is due to their choice of IPTG over aTc, and mention in the text what would happen if they used aTc as one of their main control parameters?

5.- The authors fit 11 parameters from their experimental data. Could the number of parameters be reduced by non-dimensionalization?

I also have the following minor comments:

6.- In the first paragraph of the Results section the authors say "The dichotomous response of the cell depends on the asymmetry of the repression strengths and on the production rates of each node". This makes it seem as if the bistable response requires an asymmetry in the repression strengths and production rates. I would ask the authors to rephrase this sentence.

7.- The placement of IPTG in Fig. 1B is misleading and does not correspond to the one in Fig. 1A (which I think it's the correct choice).

8.- The value of IPTG is missing in the caption of Fig. S1B. I assume it's 0; what happens then if IPTG is large? Also, what the authors mean by "previous state" in this figure seems to be what they referred to as "initial state" in the main figures. It would be best to be consistent with the notation throughout the paper.

9.- On page 6 the authors say "Analysis of the bifurcations of the system shows a scenario compatible with other TS, in which...". What other TS are they referring to?

10.- The authors introduce the concept of "critical slow down" on page 10. The common term for this effect is "critical slowing down", if I'm not wrong. It would also be nice to add a reference to this.

11.- On page 12 the authors say "this was the case even for short pulses were not the whole population did have enough time to switch". It should be "where" rather than "were", if I understood the sentence correctly.

In any case, as I said above I think this is a nice paper overall, and I would like to thank the authors for this detailed study, which as mentioned I would like to use in my teaching in the future.

Response to reviewer's comment**Reviewer #2:**

In the manuscript "Controlling spatiotemporal pattern formation in a concentration gradient with a synthetic toggle switch," the author's explore how a "classical" regulatory motif, the toggle switch, composed of two cross-repressing transcription factors, may convert continuous information provided by the gradient into discrete stripes of gene expression. Towards this end, the authors build a synthetic biology framework to understand and characterize the spatiotemporal patterning properties of the toggle switch. They build a synthetic toggle switch controllable by diffusible molecules in *Escherichia coli* and explore the "patterning" by combining quantitative measurements with a mathematical reconstruction of the system. The authors find that the toggle switch can produce robust patterns with sharp boundaries, bistability, and hysteresis in a tunable manner.

A key conclusion is that by building a mathematical representation-namely bifurcation diagrams of the system-they were able to explore the various spatial-temporal patterns observed and experimental "guide" pattern formation. A key finding was that they could characterize the switch from bistability to sigmoidal response. In sum, the methodology combining modeling with their grid-based systems and flow-cytometry was excellent and allowed the exploration of a classic network motif.

We thank the reviewer for this positive comment.

While the work is compelling, and the conclusions appear solid, the main shortcoming is that many aspects of this work are not entirely novel and may not represent a significant advance in the field. For example, a bistable genetic switch based on designable DNA-binding domains has been explored by Lebar and colleagues (Nat. Comm. 2014). Similar to the present work, these authors found non-linearity that results in epigenetic bistability. Others have also built bistable systems that not switchable, depending on the initial input conditions (Kim et al., 2006). See also Pokhilko et al., 2019.

We agree with the reviewer that the toggle built has been built and analysed previously. Indeed we write "Since then, it has been built multiple times, extensively studied and used for its memory, bistability or hysteresis properties (Padirac et al., 2012, Purcell and Lu, 2014, Chen and Arkin, 2012, Lou et al., 2010, Andrews et al., 2018, Yang et al., 2019, Nikolaev and Sontag, 2016, Zhao et al., 2015, Sokolowski et al., 2012, Lebar et al., 2014), for stochasticity fate choice (Axelrod et al., 2015, Perez-Carrasco et

al., 2016, Sekine et al., 2011, Wu et al., 2013, Lugagne et al., 2017) and to tune threshold activation (Gao et al., 2018).”

We now also added the references the reviewer pointed out and we had not yet cited (Kim et al., 2006, Pokhilko et al., 2018)

However, none of the previous experimental studies looked at the spatiotemporal patterning properties of the toggle switch. We are also the first to have a (synthetic) experimental system that allowed us to transition between the bistable and the sigmoidal regimes and thus compare the properties of the different regimes. This allowed us to reveal a trade-off between speed and precision of boundary formation. Moreover, we developed a mathematical model that was used to guide the gathering of experimental data and to understand the patterning mechanisms of the circuit. The use of dynamical system descriptions of bistable systems is not new. Nevertheless, the incorporation of the wealth of data obtained to reconstruct the underlying dynamical system and infer its corresponding bifurcations is beyond common practice.

To highlight the novelty of our research better, we added following text in the discussion of the main manuscript:

“Here, for the first time we characterize a synthetic inducible toggle switch (Figure 1) in order to explore the dynamic patterning properties of the TS gene regulatory network by combining quantitative measurements with a mathematical reconstruction of the underlying dynamical system. Our setup with an AHL gradient controlling the expression level of one of the promoters and an IPTG-tunable repression allowed us to transition between bistable and sigmoidal regimes. This in turn allowed us to show how the hysteresis, position, timing, and precision of the boundary can be controlled, revealing a trade-off between speed and precision of the boundary formation.”

Finally, while the descriptions of developmental systems are interesting, the authors should highlight the caveats of their system. Namely, the transcription within animal systems is highly dynamic, with rapid binding kinetics. For example, Sox2 dwells at specific target DNA for only ~12.0~14.6s. Could their system capture this? If yes, they should describe it.

The referee is right on pointing out the dynamical differences between prokaryotic and eukaryotic gene regulation. However, the main results of the paper establishes a relationship of the patterning dynamics of the circuit from a dissection of the bifurcation properties of the dynamical system described as a set of ODEs. Similar bifurcations have been applied to eukaryotic systems, suggesting that our approach will be insightful in patterning of developmental systems. Indeed, a population of *E. coli* cells carrying a synthetic circuit does not capture the complexity of a developing animal, e.g. we do not have chromatin remodelling. However, it is this reduced complexity that allowed us to obtain high quality data on the

patterning properties of the TS without confounding factors, and can be used in the future to establish how more complex regulatory mechanisms can provide spatiotemporal control of the patterning.

In addition, we do not believe that differences in dwell times are going to affect our predictions, since our model indeed assumes that the binding kinetics are faster compared to the production and degradation of the proteins. We have included this observation in the Discussion of the manuscript:

“However, extending such parallelisms further will require an exhaustive analysis of the different dynamical properties between our synthetic system in *E. coli* and eukaryotic systems. For instance, we expect the approach performed in this manuscript to be valid in eukaryotic systems in which binding kinetics and changes in promoter availability are faster than the patterning time, but not in situations where they are slower (Bintu et al., 2016).”

While the authors did not observe a clear difference in the precision of the boundary, it would be great to explore this further. Could they be more precise in the concentration gradients? Animal development is often not precise on the input side so could this be a regulatory mechanism?

We performed additional experiments to quantify the boundary precision difference between the sigmoidal regime at high IPTG concentrations and the bistable regime at low IPTG concentrations (new Figure EV4). Careful fluorescence intensity quantification demonstrated a difference in sharpness in the new grids (Figure EV4 A, B). Moreover, this phenomenon is obvious to the naked eye if we do not use a grid (Figure EV4 C).

We hypothesise that the high starting cell density (OD=3) used in the time course measurement was a reason why we did not observe the sharpness difference in this experiment. However, at this stage we cannot exclude that other factors that we did not carefully control (such as the agar plate thickness, agar pouring temperature) might influence the boundary precision.

We have now added Figure EV4 and adjusted the comment that we did not observe the boundary precision in the time course experiment to

“On the other hand, in this assay, we did not observe clear differences in the precision of the boundary, potentially due to the high starting cell density”.

Additionally, as stated in the previous question, we do not expect to have the same exact spatiotemporal patterning properties between animal and bacterial patterning. In particular, we anticipate that one of the main differences will come from the role of molecular fluctuations in both systems, not only in the morphogen gradient but also in other characteristics of the GRN such as chromatin remodelling, binding dynamics, or transcriptional bursting. We have included this in the Discussion:

“Differences might also come from other characteristics of the GRN, such as chromatin remodelling, binding dynamics, or transcriptional bursting (Bentovim et al., 2017, Mathur et al., 2017, Oates, 2011, Folguera-Blasco et al., 2019).”

In sum, this manuscript provides a fantastic ability to control the transition between regime modes (sigmoidal, bistable, and hysteresis). However, the authors would need to justify the conceptual advances in the field. Simply calling varying IPTG conditions a "morphogen" is not enough to justify how this differs from prior work.

We thank the reviewer for the feedback and hope we clarified the remaining concerns and she/he agrees that we have now better emphasised the novelty of our study.

Reviewer #3:

This paper elegantly combines experiments with mathematical modeling to study in a systematic manner the behavior of a synthetic toggle switch implemented in the bacterium *E. coli*. Specifically, the authors study the response of this circuit to spatial gradients of several inducers, which they interpret as morphogens. From this analysis, they extract conclusions relative to the stripe-forming ability of the circuit in response to those gradients in different parameter regimes, regarding aspects such as its robustness, controllability, and kinetics. While the circuit employed is rather standard, and the system is quite far from realistic situations displaying morphogen-driven pattern formation (in development, for instance), the systematic study presented here has value in itself, constituting almost a textbook analysis (which in fact I'll probably use in my classes in the future, given that the preprint is already publicly available in bioRxiv). There are some issues, however, that make the paper somewhat unclear, and which I would ask the authors to address before publication:

We thank the reviewer for these positive comments and the recommendations which we address below:

1.- According to the text, the inducers are added instantaneously on the corner of the hydrophobic grid, and then the system is left to relax for hours. If that is the case, the gradients are continuously decaying, and the time at which the experimental observations of Figs. 1, 2 and S1-S4 is made should play an important role. The authors state that the measurements are taken after overnight incubation, but this is too vague in my opinion. What's the exact time? What happens if that time is doubled, for instance? And why can these results be fitted to the steady state of their model? These questions should be substantially clarified in the manuscript.

We thank the reviewer for highlighting this point that we forgot to discuss. After overnight incubation, which is around 16 h (now specified in the figure captions, text and the methods), the pattern does not change anymore even after one additional day of incubation. Once the cells enter stationary phase,

they do no longer switch between the two states. This unresponsiveness in stationary phase is a well-known phenomenon of many synthetic circuits. Therefore, in our flow cytometry experiments, we took great care to avoid stationary phase as pointed out in the methods:

“Absorbance (600 nm) was monitored every 10 min to check that cells did not enter stationary phase (below 0.3 in the plate reader) as cells in stationary phase can no longer switch between the two states.” It is also one of the reasons why we do not follow the grid in Fig.4 for longer than 10 h.

And now also explained in the main text:

“We prevented the bacteria entering stationary phase by diluting them 100-fold after 5 h.”

To clarify this aspect, we added in the main text:

“We measured the bacterial fluorescence after overnight incubation (~16 h). Although the gradients decay over time, the observed patterns stayed constant even after further incubation, as the expression of the synthetic toggle switch become frozen in cells that have entered the stationary phase, a phenomenon commonly observed for bacterial synthetic circuits (Elowitz and Leibler, 2000)”

In addition, all the switching dynamics are observed as a change in density of the two clouds corresponding to both cellular states. This is a signature that the local transient relaxation to each local cellular state is faster than the switching dynamics between them and allows to characterize the system by identifying the position of these states with the steady states of the deterministic dynamical system. This has now been clarified in the main text:

“Since the transient expression profiles to both cellular states were faster than the transitions between them, we inferred the details of the dynamical system by identifying each observed cellular state with a stable steady state of the dynamical system.”

2.- How is the boundary for the gating of the flow cytometry measurements chosen? It would be helpful if this were also clarified in the text.

The gating was based on green and red positive controls (as described in the methods).

We mention this now also in the main text:

“We set gates for the red and green states with the help of positive red and green fluorescent controls.”

3.- One point that I find interesting is the difference between bistability and bimodality reported in Fig. 2A. I guess that the reason for this difference is the fact that the potential barrier between the two stable states in part of the bistable regime is too large for jumps to take place by the time the measurements are taken. In fact, the authors invoke this when comparing between the sigmoidal and bistable regimes, but I would expect that even within the bistable regime itself one could find the two behaviours, depending on the time at which the measurements are made. In fact, I would expect the two regions (bimodality and bistability) to grow more similar to each other the later the measurements are made

(with a limit given by the time at which the gradients virtually disappear due to diffusion). Am I correct? If so, I would ask the authors to clarify this in the text. On the other hand, if I'm wrong and the observations made here can be interpreted in terms of a steady state behavior, it would be nice to calculate the effective potential of the system as was previously done by Wang et al in 2010 (DOI 10.1016/j.bpj.2010.03.058).

The reviewer is completely right with this remark, bimodality should be observed inside all the bistable zone if we waited long enough. We have now clarified this difference in the main text:

“In particular, the bimodal profile is only observed close to the bifurcation, where the time scale of noise induced transitions between cellular states is smaller than the duration of our observation time (10 h) (Perez-Carrasco et al., 2016).”

Unfortunately, for bacterial circuits, the window of time to observe a transition is limited to the time that the cells are in the exponential (log) growth phase (before reaching the stationary phase) – please see the answer to your first question. In order to avoid entering stationary phase, we therefore periodically diluted the samples. As with our equipment it was not possible to automate this process, we limited the time of observation to 10 h. This not only reduces the size of the observed bimodal region but also impedes to characterize accurately the potential of the system by assuming long time steady state probability distributions as in Wang et al.

4.- This system has three different inductors, namely AHL, IPTG and aTc. Given the symmetry of the mutual inhibition switch, the natural two parameters to focus on would be, in my opinion, IPTG and aTc, whereas AHL would be a general inducer of the circuit. However, the authors decided to focus mainly on AHL and IPTG, and this asymmetry leads to the asymmetry reported on page 6: "The transition from red to green is less affected by the concentrations of IPTG than the transition from green to red, because of the asymmetry of the network". Can the authors confirm whether the need to focus on initial green states is due to their choice of IPTG over aTc, and mention in the text what would happen if they used aTc as one of their main control parameters?

The statement on the symmetry of the system is correct. In particular, the asymmetry we reported on page 6 would have been lower, if we used aTc-IPTG instead of IPTG-AHL.

On the other hand, we favoured to choose AHL-IPTG over aTc-IPTG because we aimed to have a TS which interprets a “morphogen” gradient (AHL) that acts on the expression level of the target by direct promoter interaction, similar to known morphogens like bicoid (Chen et al. 2012 10.1016/j.cell.2012.03.018) or Sonic Hedgehog (Balaskas et al. 2012), rather than changing the repression strength of the repressor in the toggle switch.

In order to answer to what would happen if we used aTc, we can analyse and compare the figures 1C-E and S1D, combined here in the following figure:

Please note that AHL and aTc have opposite effects: AHL activates the red node and aTc represses the repression of TetR, which increases the expression of the green node and thus represses the red node. For the sake of comparison, we are showing the gradients of AHL and aTc in opposite directions. If we compare the patterns for an initial green state, we notice a similar behaviour in both cases. A wide green area where IPTG is absent and only a red area in presence (AHL) or absence (aTc) of the inducer. However, if we start from the red state, we observe a smaller green area in the aTc-IPTG case compared to the AHL-IPTG case. This shrunken green area implies a difference in the hysteresis behaviour. Hence, for the aTc gradient, the bistability at intermediate concentration of inducer is possible and even wider than for AHL. In addition, the irreversible behaviour is also possible. For example, the green state at high IPTG concentration along the aTc gradient is only accessible if the system starts from the green initial state. Furthermore, this irreversible regime can be at the four borders of the grid. By contrast to the AHL gradient, the sigmoidal regime (without hysteresis) is not present for an aTc gradient in our system (although we do not exclude that this would be possible by tuning the parameter of the system). Thus, when using aTc-IPTG, we would not have been able to compare the properties of the sigmoidal and bistable regimes and we would not have revealed the trade-off between speed and precision of boundary formation.

To explain the reader our choice, we extended our manuscript with the following text in the results: “We focused on AHL-IPTG because we aimed to have a TS which interprets a “morphogen” gradient (AHL) that acts on the expression level of the target by direct promoter interaction, similar to known morphogens like bicoid (Chen et al., 2012) or Sonic Hedgehog (Balaskas et al., 2012). Moreover, this combination allowed us to observe a cusp bifurcation as we will describe below.”

and in the conclusions:

“Importantly, this was enabled by the choice of focusing on AHL and IPTG gradients. While combining aTc and IPTG gradients would have reduced the asymmetry of the system, we would not have been able to reach the sigmoidal regime (Figure EV1).”

5.- The authors fit 11 parameters from their experimental data. Could the number of parameters be reduced by non-dimensionalization?

The equations used to infer the parameters of the system make use of an irreducible set of non-dimensionalized identifiable parameters. The original set of equations has 15 parameters. The non-dimensionalization is achieved by defining the effective production parameters $\tilde{\beta}_X = \omega_X \beta_X / \delta_X$, and $\tilde{\beta}_Y = \omega_Y \beta_Y / \delta_Y$ that incorporate the time scale of the system through the degradation rates (δ_X and δ_Y) and the fluorescence intensity scales (ω_X and ω_Y). This is now clarified in the methods:

“Thus, the parametrization of the problem is reduced to the inference of 11 irreducible identifiable parameters $\theta = \{\alpha_X, \alpha_Y, \tilde{\beta}_X, \tilde{\beta}_Y, K_{TetR}, K_{LacI}, K_{AHL}, K_{IPTG}, n_{TetR}, n_{LacI}, n_{AHL}\}$, where $\tilde{\beta}_X$ and $\tilde{\beta}_Y$ are the non-dimensionalizing production rates summarizing the parameter products $\tilde{\beta}_X = \omega_X \beta_X / \delta_X$, and $\tilde{\beta}_Y = \omega_Y \beta_Y / \delta_Y$ ”

I also have the following minor comments:

6.- In the first paragraph of the Results section the authors say "The dichotomous response of the cell depends on the asymmetry of the repression strengths and on the production rates of each node". This makes it seem as if the bistable response requires an asymmetry in the repression strengths and production rates. I would ask the authors to rephrase this sentence.

We rephrased this and combined it with the following sentence so that it reads now: “An array of cells under a concentration gradient in charge of controlling either the repression strength or the production rate of one of the nodes will generate a binary spatial pattern.”

7.- The placement of IPTG in Fig. 1B is misleading and does not correspond to the one in Fig. 1A (which I think it's the correct choice).

We thank the reviewer for noticing this mistake. Indeed, IPTG does not repress LacI expression, but reduces the repression strength of LacI, by reducing the number of LacI proteins that can bind to the pLuxLac promoter. We changed figure 1A accordingly.

8.- The value of IPTG is missing in the caption of Fig. S1B. I assume it's 0; what happens then if IPTG is large? Also, what the authors mean by "previous state" in this figure seems to be what they referred

to as "initial state" in the main figures. It would be best to be consistent with the notation throughout the paper.

That is correct, the values of IPTG for Fig S1B (now renamed Expanded view 1B) was 0. This has been added to the figure legend.

We now also performed the grid assay with an IPTG concentration of 10 mM (Expanded view 1C). IPTG allows the red area to form in the presence of AHL and in the absence of aTc.

The reviewer is right with its last suggestion, thereof we changed "previous state" to "initial state" in FIG S1 (now renamed Expanded view 1)

9.- On page 6 the authors say "Analysis of the bifurcations of the system shows a scenario compatible with other TS, in which...". What other TS are they referring to?

This statement makes reference to the majority of literature analysing bifurcation diagrams of genetic toggle switches. We have added some references to make it clearer:

"Analysis of the bifurcations of the system shows a scenario compatible with other TS, in which a continuous variation of AHL or IPTG can change the number of stable states via saddle-node bifurcations (Gardner et al., 2000, Perez-Carrasco et al., 2016, Guantes and Poyatos, 2008)."

10.- The authors introduce the concept of "critical slow down" on page 10. The common term for this effect is "critical slowing down", if I'm not wrong. It would also be nice to add a reference to this.

The reviewer is correct "critical slowing down", is the term for this effect. We changed it and added two related references:

"In addition, integrating the diffusion of the inducer with the dynamical properties of the bifurcations of the system can shed light on the different patterning dynamics observed. In particular, consistent with our flow cytometry data (Figure 3), the model suggests that the switching slows down close to the saddle node bifurcation, a phenomenon called critical slowing down (Perez-Carrasco et al., 2016, Narula et al., 2013)."

11.- On page 12 the authors say "this was the case even for short pulses were not the whole population did have enough time to switch". It should be "where" rather than "were", if I understood the sentence correctly.

We thank the reviewer for noticing this mistake. We corrected "were" to "where".

In any case, as I said above I think this is a nice paper overall, and I would like to thank the authors for this detailed study, which as mentioned I would like to use in my teaching in the future.

We thank the reviewer for this positive feedback and we would be pleased if our study will be used in teaching.

Dear Prof Schaerli,

Thank you for sending us your revised manuscript. We have now heard back from the two reviewers who agreed to evaluate your manuscript. You will see from the comments below that both reviewers are overall positive about the revision and support publication of the article in *Molecular Systems Biology*. I am pleased to inform you that your manuscript will be accepted in principle pending the following essential amendments:

1. Please reduce the keyword number to 5.
2. Please change "Declaration of interest" into "Conflict of interest".
3. Datasets: Please update the nomenclature of the Dataset 1& 2 into Dataset EV1&2, and update the callout in the main manuscript accordingly. Further, please provide legends for these datasets and insert the legends into each zip file in a docx or txt file.
4. Checklist: please enter the correspondence author name, journal name, manuscript number in the checklist.
5. I have slightly modified the synopsis text, can you please let me know if it is fine as it is or if you would like to introduce further modifications?

Synopsis:

Toggle switch is a common subnetwork of gene regulatory networks in charge of pattern formation. This study combines a synthetic biology framework and mathematical modeling to characterize the spatiotemporal properties of toggle switch in *E.coli*.

- A synthetic toggle switch network in *E. coli* interprets a signal concentration gradient into bistable and hysteretic spatial patterns.
- Combining quantitative measurements with a mathematical model allows reconstructing the underlying bifurcation diagram.
- Modulating the repression strength of the mutual repressing nodes allows to control the hysteresis, position, timing, and precision of the boundary.

When you resubmit your manuscript, please download our CHECKLIST (<http://bit.ly/EMBOPressAuthorChecklist>) and include the completed form in your submission. *Please note* that the Author Checklist will be published alongside the paper as part of the transparent process (<https://www.embopress.org/page/journal/17444292/authorguide#transparentprocess>)

Click on the link below to submit your revised paper.

Link Not Available

Yours sincerely,

Jingyi Hou
Editor
Molecular Systems Biology

If you do choose to resubmit, please click on the link below to submit the revision online before 25th May 2020.

Link Not Available

IMPORTANT: When you send your revision, we will require the following items:

1. the manuscript text in LaTeX, RTF or MS Word format
2. a letter with a detailed description of the changes made in response to the referees. Please specify clearly the exact places in the text (pages and paragraphs) where each change has been made in response to each specific comment given
3. three to four 'bullet points' highlighting the main findings of your study
4. a short 'blurb' text summarizing in two sentences the study (max. 250 characters)
5. a 'thumbnail image' (550px width and max 400px height, Illustrator, PowerPoint or jpeg format), which can be used as 'visual title' for the synopsis section of your paper.
6. Please include an author contributions statement after the Acknowledgements section (see <https://www.embopress.org/page/journal/17444292/authorguide#manuscriptpreparation>)
7. Please complete the CHECKLIST available at (<http://bit.ly/EMBOPressAuthorChecklist>). Please note that the Author Checklist will be published alongside the paper as part of the transparent process (<https://www.embopress.org/page/journal/17444292/authorguide#transparentprocess>).
8. Please note that corresponding authors are required to supply an ORCID ID for their name upon submission of a revised manuscript (EMBO Press signed a joint statement to encourage ORCID adoption) (<https://www.embopress.org/page/journal/17444292/authorguide#editorialprocess>).

Currently, our records indicate that the ORCID for your account is 0000-0002-9083-7343.

Link Not Available

The system will prompt you to fill in your funding and payment information. This will allow Wiley to send you a quote for the article processing charge (APC) in case of acceptance. This quote takes into account any reduction or fee waivers that you may be eligible for. Authors do not need to pay

any fees before their manuscript is accepted and transferred to the publisher.

*** PLEASE NOTE *** As part of the EMBO Press transparent editorial process initiative (see our Editorial at <http://dx.doi.org/10.1038/msb.2010.72> , Molecular Systems Biology will publish online a Review Process File to accompany accepted manuscripts. When preparing your letter of response, please be aware that in the event of acceptance, your cover letter/point-by-point document will be included as part of this File, which will be available to the scientific community. More information about this initiative is available in our Instructions to Authors. If you have any questions about this initiative, please contact the editorial office (msb@embo.org).

Reviewer #2:

Overall, I feel that the authors have addressed the points brought up by the reviewers.

Reviewer #3:

The authors have addressed satisfactorily my comments.

The Authors have made the requested editorial changes.

Corresponding Author Name: Rubén Perez Carrasco and Yolanda Schaeferli

Manuscript Number: MSB-19-9361